# Fast and Low-Cost Genomic Foundation Models via Outlier Removal

**Haozheng Luo** [* 1]  **Chenghao Qiu** [* 2]  **Maojiang Su** [1]  **Zhihan Zhou** [1]  **Zoe Mehta** [3]  **Guo Ye** [1]  **Jerry Yao-Chieh Hu** [1]  **Han Liu** [1]

## Abstract

To address the challenge of scarce computational resources in genomic modeling, we introduce **GERM**, a genomic foundation model with strong compression performance and fast adaptability. GERM improves upon models like DNABERT-2 by eliminating outliers that hinder low-rank adaptation and post-training quantization, enhancing both efficiency and robustness. We replace the vanilla attention layer with an outlier-free mechanism inspired by associative memory models. By removing outliers during both pre-training and fine-tuning, this approach accelerates adaptation, reduces computational costs, and enhances quantization robustness within acceptable loss margins. Additionally, we propose **GERM-T**, a strategy that employs small-step continual learning within the outlier-free framework, leveraging original checkpoints to avoid retraining from scratch. Empirically, GERM improves fine-tuning performance by 37.98% and quantization by 64.34% over the baseline model. It also reduces average kurtosis by 92.14% and maximum infinity norm by 82.77%. Compared to leading methods, GERM consistently delivers superior performance, offering a practical solution for genomic modeling in resource-constrained settings. Code is available at https://github.com/MAGICS-LAB/GERM.

## 1 Introduction

We introduce a novel model named **GERM** by utilizing outlier-free Hopfield layer (Hu et al., 2024a) to replace traditional attention layer (Vaswani et al., 2017). GERM offers a quantization-friendly and rapidly adaptable DNA genomic foundation model (GFM), making it ideal for deployment and fine-tuning on resource-constrained devices.

Existing GFMs, such as DNABERT-2 (Zhou et al., 2024) and GenomeOcean (Zhou et al., 2025b), achieve state-of-the-art performance on various genomics tasks. However, many GFM users include not only professional computational researchers but also researchers from traditional biomedical labs, who often operate on resource-constrained platforms such as mobile phones, edge devices, and IoT systems. The large size and high computational cost of these models make them challenging to use in such devices. Also, if researchers require the model to adapt to new tasks, it should be able to be fine-tuned on those tasks without demanding substantial computational resources. Efficient methods such as low-rank adaptation fine-tuning (e.g., LoRA (Hu et al., 2022)) and post-training quantization (e.g., SmoothQuant (Xiao et al., 2023)) help reduce training and inference costs for GFMs. However, directly applying these techniques to original models without modification leads to huge performance drops. This results from outlier values in the model's attention mechanisms, inherited from pretrained models (Clark et al., 2019; Kovaleva et al., 2019). Prior studies (Hu et al., 2024a; Bondarenko et al., 2024; Clark et al., 2019) show that transformer-based models often direct attention toward less useful tokens, referred to as outliers. These outliers cause inefficiencies that reduce the overall model performance. Additional studies (Wu et al., 2024c; Huang et al., 2024; Hu et al., 2025) reveal that low-rank adaptation worsens the outlier issue. Outliers from both pretrained models and low-rank adaptation fine-tuning processes distort outputs and lower accuracy.

To address inefficiencies caused by outliers in transformer-based genomic foundation models, GERM draws inspiration from associative memory models (Hu et al., 2024a;b; 2023; Xu et al., 2024; Wu et al., 2024a;b; Ramsauer et al., 2021). We replace the standard transformer attention mechanisms with an outlier-free attention layer proposed by Hu et al. (2024a), which detects and removes outliers occurring dur-

---

[*]Equal contribution [1]Northwestern University [2]Tianjin University [3]Vernon Hills High School. Correspondence to: Haozheng Luo <hluo@u.northwestern.edu>, Chenghao Qiu <q1320460765@tju.edu.cn>, Maojiang Su <maojiangsu2030@u.northwestern.edu>, Zhihan Zhou <zhihanzhou2020@u.northwestern.edu>, Zoe Mehta <zoe.mehta@vhhscougars.org>, Guo Ye <guoye2018@u.northwestern.edu>, Jerry Yao-Chieh Hu <jhu@u.northwestern.edu>, Han Liu <hanliu@northwestern.edu>.

*Proceedings of the 42nd International Conference on Machine Learning*, Vancouver, Canada. PMLR 267, 2025. Copyright 2025 by the author(s).

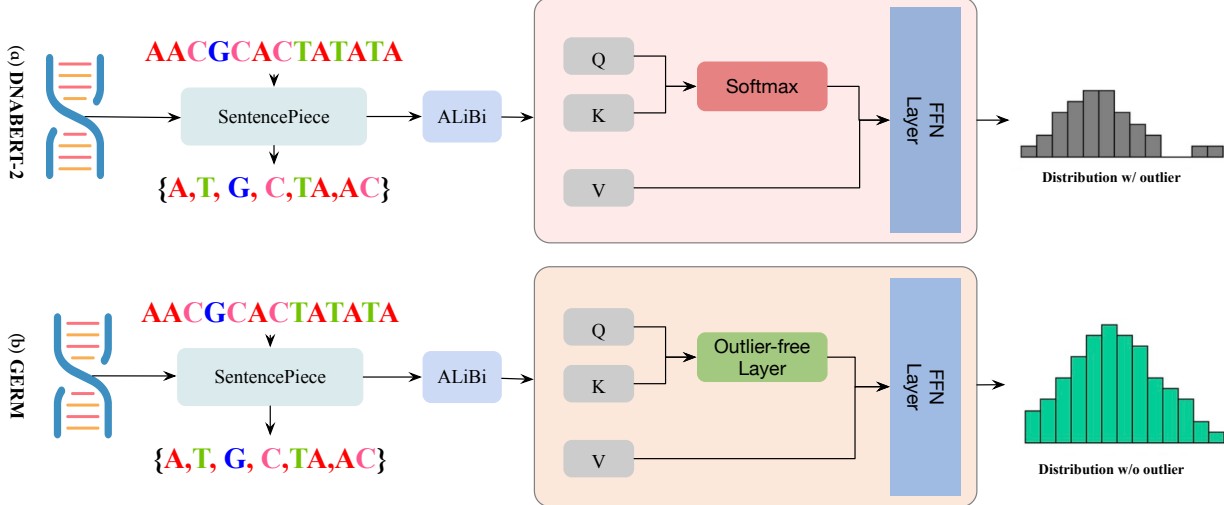

Figure 1: **Structural Comparison of DNABERT-2 and GERM Models.** This diagram illustrates the differences in processing pipelines between DNABERT-2 and GERM. Both DNABERT-2 and GERM use the SciencePiece tokenizer with BPE for tokenization. Following that, both models employ ALiBi for positional encoding in the embedding layer. However, as shown in (a), DNABERT-2's transformer architecture outputs the outliers. We propose replacing the vanilla Softmaxwith an outlier-free layer. In (b), the output of the attention mechanism removes outliers from the original output.

ing pretraining and low-rank adaptation (LoRA).

This outlier mitigation in GERM results in a "triple win" for genomic foundation models: faster low-rank adaptation, reduced computational demands, and more reliable post-training quantization. On resource-constrained devices, incorporating fine-tuning techniques such as QLoRA (Dettmers et al., 2024a) and quantization methods like OmniQuant (Shao et al., 2024) enables efficient fine-tuning and inference with minimal performance degradation. This significantly enhances its accessibility and usability, promoting broader deployment without specialized hardware.

Additionally, addressing the limitations of (Hu et al., 2024a), particularly its resource-heavy training from scratch, we introduce GERM-T. GERM-T adds an outlier-free layer to existing GFM and uses small-step continual training to efficiently achieve near-optimal performance.

**Contributions.** We propose **GERM**, an outlier-free GFM with enhanced quantization robustness and rapid low-rank adaptation. Our contributions are as follows:

- We propose an outlier-free model structure to address and mitigate outliers introduced by pretrained models and low-rank adaptation. This approach enables rapid low-rank adaptation and robust post-training quantization, significantly enhancing the overall performance of the quantized model and model finetuning. Notably, our model fine-tunes DNABERT in just 5 minutes on a single NVIDIA GeForce RTX 2080 Ti GPU.

- Methodologically, we replace the standard transformer attention mechanism in the GFM with an outlier-free layer to enhance the model's ability to handle and mitigate outliers during pretraining and fine-tuning. Additionally, we introduce a continual learning strategy as a compromise version to avoid retraining the model from scratch. This strategy ensures suboptimal performance in terms of model quantization robustness and low-rank adaptation.

- Experimentally, We evaluate the performance and efficiency of our method using the existing DNABERT-2 model (Zhou et al., 2024) structure. Additionally, we benchmark it against the state-of-the-art low-rank adaptation methods and post-training quantization techniques. Compared to the standard framework, the proposed framework achieves average performance improvements of **37.98%** in finetuning and **64.34%** in quantization, respectively. Additionally, GERM shows a reduction of **92.14%** in the average kurtosis and **82.77%** in the maximum infinity norm on average.

### Related Work

**Quantization.** Considering the quantized object, exiting foundation models (FMs) quantization can be classified into two fields: weight-only quantization and weight-activation quantization. For **weight-only quantization**, prior studies focus on converting weights to low-bit values. For instance, GPTQ (Frantar et al., 2023) uses block-wise reconstruction for 3/4-bit quantization. SpQR (Dettmers et al., 2024b), OWQ (Lee et al., 2024), and AWQ (Lin et al., 2024) emphasize the significance of weights tied to higher-magnitude activations. Therefore, SpQR and OWQ employ mixed-precision quantization to safeguard vital weights, while AWQ opts for channel-wise scaling to avoid

mixed-precision's hardware inefficiency. QLoRA (Dettmers et al., 2024a), LoftQ (Li et al., 2023) and QUIP (Chee et al., 2023) restore the capabilities of the quantized model through parameter-efficient fine-tuning. For **weight-activation quantization**, prior studies compress both weights and activations. SmoothQuant (Xiao et al., 2023), LLM.int8() (Dettmers et al., 2022), and Outlier Suppression (Wei et al., 2022) achieve W8A8 quantization by managing activation outliers. LLM.int8() uses mixed-precision decomposition, while the other two employ channel-wise scaling. Furthermore, Outlier Suppression+ (Wei et al., 2023) adds channel-wise shifting to drive W6A6 quantization. In comparison to other quantization approaches, including prior works (Wei et al., 2023; Xiao et al., 2023) that address the outlier issue during quantization, the outlier-free layer in GERM is more effective at managing outliers within the model's attention mechanism. It provides GERM with a unique advantage in terms of quantization robustness.

**Outlier Values in Quantization.** Numerous studies (Hu et al., 2024a; Ma et al., 2024; Heo et al., 2024; Puccetti et al., 2022; Kovaleva et al., 2021; Bondarenko et al., 2021; Luo et al., 2021) observe outlier values in the transformer-based language models such as BERT (Devlin et al., 2019) and early GPT (Radford et al., 2019) models. Since the advent of FMs (Zhou et al., 2024; 2025a; Zhang et al., 2022; Brown et al., 2020) root in the GPT and BERT, recent studies by Xiao et al. (2023); Ahmadian et al. (2023); Dettmers et al. (2022) tackle the existence of outlier values in FMs. According to them, these outliers exhibit a large magnitude of values at the shared dimensions of hidden states across tokens. More recently, Bondarenko et al. (2024); Sun et al. (2024); Hu et al. (2024a) explain that the outliers attribute to the vertical pattern in the attention mechanism (Xiao et al., 2024; Kovaleva et al., 2019), influencing the performance of FMs. In particular, Sun et al. (2024) claim a different type of outlier existing in the hidden states of specific tokens. However, most of these studies concentrate on language and vision models, leaving the impact of outliers on genomic foundation models largely unexplored. Additionally, methods like Hu et al. (2024a) require training from scratch to eliminate outliers, which is computationally expensive.

**Genomic Foundation Model.** The majority of genomic foundation models (GFMs) use transformers to model sequence dependencies, similar to BERT (Devlin et al., 2019) and GPT (Brown et al., 2020) in NLP. Specifically, DNABERT (Ji et al., 2021) and DNABERT-2 (Zhou et al., 2024) leverage transformers for DNA sequence analysis by employing masked language modeling and fine-tuning for biological tasks. In addition, Nucleotide Transformer (Dalla-Torre et al., 2024) excels at molecular phenotype prediction and variant prioritization, while HyenaDNA (Nguyen et al., 2024b) is optimized for modeling long-range genomic

dependencies. Furthermore, GenomeOcean (Zhou et al., 2025b) provides an efficient 4-billion-parameter genome foundation model for diverse, context-aware DNA sequence generation. However, these models demand significant computational resources and lack robustness to quantization, rendering them unsuitable for deployment on resource-constrained devices. Specifically, GenomeOcean utilizes 64 NVIDIA A100 80G GPUs over a span of 14 days for training. This limits accessibility for research labs with limited computational capacity. More recently, Evo (Nguyen et al., 2024a), a generative genomic model, integrating Transformer and Hyena operator to efficiently capture long-range dependencies in genomic sequences, achieving a context window of 131k nucleotides. Furthermore, Evo uniquely bridges bridges the DNA-RNA-protein central dogma via cross-modal inference without task-specific supervision.

## 2 GERM

This section introduces the proposed method, which comprises the outlier-free architecture, small-step continual learning, and the DNA genomic foundation model (GFM). The outlier-free architecture is designed to mitigate challenges posed by outliers during the model fine-tuning process. Meanwhile, the small-step continual learning technique extends the training process using smaller learning steps after the initial training, aiming to address and mitigate the outliers present in the original model checkpoints.

In our study, we develop the GFM framework to train DNA sequence-based genomic foundation models that employ Transformer-based architectures such as DNABERT (Ji et al., 2021) and Nucleotide Transformer (Dalla-Torre et al., 2024). These models use DNA tokenization and Transformer attention mechanisms, making them well-suited for integrating techniques like LoRA and the outlier-free mechanisms proposed in our approach. Alternatively, models like HyenaDNA (Nguyen et al., 2024b) and Caduceus (Schiff et al., 2024) utilize different architectures, such as convolutional layers or the Mamba architecture. While these models introduce novel features, they are not currently the most widely adopted in genomic modeling and require further research. Therefore, we adopt DNABERT-2 as the baseline in this paper, as it best represents the Transformer-based DNA GFMs central to our study. The proposed outlier-free architecture of GERM is illustrated in Figure 1.

**Outliers Challenge in Transformer Architecture.** Previous studies (Clark et al., 2019; Kovaleva et al., 2019) demonstrate that structural elements, including delimiters and sentence boundaries, attract unexpectedly high attention weights in BERT's attention mechanism. Consequently, these tokens dominate the attention mechanism, overshadowing more informative tokens. Further anal-

ysis by Kobayashi et al. (2020) show that tokens with smaller value vector magnitudes paradoxically tend to obtain greater attention weights. These phenomena indicate that transformer-based models may focus on less relevant information, leading to inefficient processing. Studies by Hu et al. (2024a); Bondarenko et al. (2024) highlight the underlying cause of the outlier challenge in transformer-based models, proposing that transformers do not require updates when the attention inputs are sufficiently informative. However, the normalization nature of the $\mathrm{Softmax}$ function forces non-zero attention weights even for irrelevant tokens, creating numerical instability. Such outliers distort gradient updates and hinder model performance. Additionally, this issue increases computational and memory demands during training and results in significant performance degradation after model quantization. Consequently, implementing a strategy to address outliers during both the pretraining and fine-tuning stages is crucial. Numerous studies address the outlier problem across different model stages, including pre-training (Hu et al., 2024a), fine-tuning (Hu et al., 2025), and inference (Bondarenko et al., 2024; Xiao et al., 2023). In our study, we extend the work of Hu et al. (2024a) by utilizing the memory-associated retrieval dynamics function $\mathrm{Softmax}_1$. This function is defined as

$$\mathrm{Softmax}_1(S) := \frac{\exp(S)}{1 + \sum_{i=1}^{L} \exp(S_i)},$$

where $S$ is the input to the activation function. This approach addresses outlier problems in GFMs. Additionally, we provide a theoretical analysis of the expressive guarantee of low-rank adaptation for transformer-based GFMs with $\mathrm{Softmax}_1$ in Appendix A.

**Small-step Continual Learning.** The outlier removal technique introduced in `OutEffHop` (Hu et al., 2024a) is highly effective in reducing the impact of outliers during model pretraining. However, a major limitation of this approach is the need to retrain the model from scratch, which is a significant challenge for large-scale models like GFMs due to the extensive time and computational resources required. To address this issue, we suggest a small-step continual learning approach as a compromise to the existing GERM structure, called GERM-T. It involves resuming training with an outlier-free model structure after the initial training phase to address and mitigate outliers in the original model checkpoints. This approach aims to lower the computational cost and time needed for retraining while still optimizing performance. Although this small-step continual learning technique may not be as effective as full retraining, it offers a more efficient and cost-effective solution for outlier removal in GFMs. For users with limited computational resources who cannot train a model from scratch and rely on 8-bit or 6-bit quantization during inference, GERM-T offers a viable compromise strategy.

**DNA Genomic Foundation Model.** We implement a simple yet effective design for the DNA genomic foundation model (GFM) following DNABERT-2 (Zhou et al., 2024). Initially, we employ SentencePiece (Kudo & Richardson, 2018) with Byte Pair Encoding (BPE) (Sennrich et al., 2016), a subword tokenization method, to process DNA sequences. SentencePiece is particularly effective for handling the large number of unique tokens present in DNA without assuming any pre-tokenization, such as k-mer segmentation (Chor et al., 2009). SentencePiece with BPE is used in natural language processing for word segmentation and learns a fixed-sized vocabulary of variable-length tokens based on character co-occurrence frequencies. Due to the significant difference between natural language and DNA sequences, the vocabulary sizes used in the NLP domain (Zhang et al., 2022; Radford et al., 2019; Kenton & Toutanova, 2019) are not suitable for DNA sequences. In our study, we set vocabulary size to 4096, as it best balances model performance with computational efficiency among candidates.

We then adopt the BERT architecture (Kenton & Toutanova, 2019) to train our GFM on DNA sequences, with several modifications to better accommodate the unique characteristics of the DNA data. Standard positional encoding methods, such as Rotary Positional Encoding (Su et al., 2024) and Sinusoidal Positional Encoding (Vaswani et al., 2017), face limitations when applied to sequences longer than those encountered during training due to their inherent input length restrictions. To address these limitations, we employ the Attention with Linear Biases (ALiBi) method (Press et al., 2022), as it is more robust to variations in sequence length and can handle longer sequences compared to traditional positional encoding methods. Instead of adding positional embeddings to the input, ALiBi introduces linear biases into the attention mechanism, allowing the model to learn positional information inherently from the input sequence. Specifically, let $q_i \in \mathbb{R}^d$ represent the query vector for the $i$-th token in a sequence of length $L$, and $K \in \mathbb{R}^{L \times d}$ denote the key matrix for all tokens. The attention score for query $q_i$ is computed as: $\mathrm{Softmax}(q_i K^\top + m \times [-(i-1), \ldots, -1, 0, -1, \ldots, -(L-1-i)])$, where $m$ is a fixed scalar. ALiBi uses a geometric sequence of different $m$ values for each attention head, allowing model to learn positional information from the input sequence itself. By replacing learned positional embeddings with ALiBi, GERM can process arbitrarily long sequences during fine-tuning and inference, despite being pre-trained on relatively shorter sequences.

## 3  Experimental Studies

In this section, we perform a series of experiments to demonstrate the effectiveness of our proposed method. In particular, we compare the performance of our method with DNABERT-2 detailed in (Zhou et al., 2024).

Table 1: **Comparing GERM and GERM-T with DNABERT-2 in a Post-Training Quantisation (PTQ) setting.** We perform experiments on GERM with baseline models using four quantization methods (Traditional W8A8, SmoothQuant, Outlier Suppression, OmniQuant) across three quantization configurations (Weight-8bit-Activation-8bit (W8A8), Weight-6bit-Activation-6bit (W6A6), and Weight-4bit-Activation-4bit (W4A4)). The evaluation metrics include the Matthews Correlation Coefficient (MCC), the difference in MCC (Delta MCC) compared to the official DNABERT-2 checkpoint, the *average kurtosis*, and the *maximum infinity norm* $\|\mathbf{x}\|_\infty$ for outlier values at FP16. The best results are highlighted in bold, while the second-best results are underlined. In most configurations, GERM demonstrates superior fine-tuning performance compared to DNABERT-2.

| Model | #Bits | Quantization Method | MCC (↑) | Delta MCC (↓) | Avg Performance Drop (↓) | Avg. Kurtosis (↓) | Max inf. norm (↓) |
|---|---|---|---|---|---|---|---|
| Official | 16W/16A | - | 66.11 | - | - | 39.68 | 53.61 |
| DNABERT-2 | 16W/16A | - | 59.11 | 7.00 | - | 270.90 | 61.64 |
| | 8W/8A | - | 33.60±0.41 | 32.51 | 43.81% | | |
| | 8W/8A | SmoothQuant | 36.51±0.02 | 45.37 | 38.63% | | |
| | 6W/6A | SmoothQuant | 20.74±0.04 | 45.37 | 66.18% | | |
| | 4W/4A | SmoothQuant | -1.03±0.06 | 67.06 | 101.24% | | |
| | 8W/8A | Outlier | 25.26±0.02 | 40.85 | 57.60% | | |
| | 6W/6A | Outlier | 27.84±0.28 | 38.27 | 52.71% | | |
| | 8W/8A | OmniQuant | 49.92±0.05 | 16.19 | 15.76% | | |
| | 6W/6A | OmniQuant | 48.47±0.14 | 17.64 | 18.61% | | |
| | 4W/4A | OmniQuant | 2.94±0.19 | 63.17 | 94.78% | | |
| GERM | 16W/16A | - | 59.73 | 6.38 | - | **21.29** | **10.62** |
| | 8W/8A | - | 57.30±0.08 | 8.81 | **3.77%** | | |
| | 8W/8A | SmoothQuant | 56.65±0.15 | 9.46 | 4.82% | | |
| | 6W/6A | SmoothQuant | 56.48±0.07 | 9.63 | **5.45%** | | |
| | 4W/4A | SmoothQuant | 20.05±0.00 | 46.06 | **69.44%** | | |
| | 8W/8A | Outlier | 45.87±0.08 | 20.24 | **25.23%** | | |
| | 6W/6A | Outlier | 40.57±0.56 | 25.54 | 36.27% | | |
| | 8W/8A | OmniQuant | 55.99±0.09 | 10.12 | 5.95% | | |
| | 6W/6A | OmniQuant | 55.70±0.03 | 10.41 | **6.41%** | | |
| | 4W/4A | OmniQuant | 49.42±0.00 | 16.69 | **17.17%** | | |
| GERM-T | 16W/16A | - | 59.30 | 6.81 | - | 251.40 | 28.49 |
| | 8W/8A | - | 38.38±0.15 | 27.73 | 35.27% | | |
| | 8W/8A | SmoothQuant | 57.52±0.00 | 8.59 | **3.01%** | | |
| | 6W/6A | SmoothQuant | 30.34±0.04 | 35.77 | 48.83% | | |
| | 4W/4A | SmoothQuant | 0.22±0.00 | 65.89 | 99.63% | | |
| | 8W/8A | Outlier | 42.57±0.05 | 23.54 | 28.31% | | |
| | 6W/6A | Outlier | 46.02±0.06 | 20.06 | **22.34%** | | |
| | 8W/8A | OmniQuant | 56.80±0.12 | 9.31 | **4.21%** | | |
| | 6W/6A | OmniQuant | 55.41±0.00 | 10.71 | 6.57% | | |
| | 4W/4A | OmniQuant | 3.86±0.00 | 62.25 | 93.49% | | |

**Models.** Following Zhou et al. (2024), we validate our strategy with DNABERT-2 model. we adopt the DNABERT-2 model of size 117 million parameters[1]. We pretrain this model with the masked language modeling (MLM) technique, following the original DNABERT-2 (Zhou et al., 2024). Each model trained from scratch undergoes a total of 200K training steps. For models utilizing small-step continual learning, we initially train the model from scratch using the DNABERT-2 architecture, followed by continual learning with the outlier-free structure for the remaining steps. In our experiment, we use the 40K continual learning steps model as the representative example of GERM-T to compare against DNABERT-2 and GERM.

**Datasets.** We utilize 27 datasets spanning 7 tasks and 4 species, as outlined in (Zhou et al., 2024). As shown in Appendix B.3, most downstream tasks in GFMs are classification tasks. Consequently, the datasets are designed

---

[1] https://huggingface.co/zhihan1996

Table 2: **Comparing GERM and GERM-T with DNABERT-2 in a Low-Rank Adaptation Setting.** We perform experiments on GERM with baseline models across three Low-Rank Adaptation methods (LoRA, QLoRA, LoftQ). The evaluation metrics include the Matthews Correlation Coefficient (MCC), the Delta MCC performance difference relative to the official DNABERT-2 checkpoint, the *average kurtosis*, and the *maximum infinity norm* $\|\mathbf{x}\|_\infty$ for outlier values. Additionally, we measure the average performance drop after low-rank adaptation to evaluate the efficiency of GERM in this setting. The best results are highlighted in bold, while the second-best results are underlined. In most configurations, GERM demonstrates superior fine-tuning performance compared to DNABERT-2. The GERM demonstrates an average performance improvement of **37.98%** compared to the DNABERT-2 model.

| Models | Low-Rank Adaptation Method | MCC ($\uparrow$) | Delta MCC different ($\downarrow$) | Avg Performance Drop ($\downarrow$) | Avg. kurtosis($\downarrow$) | Max inf. norm($\downarrow$) |
|---|---|---|---|---|---|---|
| DNA BERT-2 | Full | 59.11 | 7.00 | - | 270.90 | 61.41 |
| | LoRA | 50.91±1.67 | 15.2 | 13.87% | - | 219.20 |
| | QLoRA | 50.65±0.13 | 15.46 | 14.31% | 292.85 | 53.91 |
| | LoftQ | 50.76±0.06 | 15.31 | 14.05% | 299.18 | 54.18 |
| GERM | Full | 59.73 | 6.38 | - | **21.29** | **10.62** |
| | LoRA | 57.27±0.70 | 8.84 | **4.12%** | - | **19.41** |
| | QLoRA | 53.16±0.21 | 12.95 | **10.99%** | 34.29 | **27.27** |
| | LoftQ | 53.11±0.08 | 13.00 | **11.08%** | 33.02 | **27.41** |
| GERM-T | Full | 59.30 | 6.81 | - | 251.40 | 28.49 |
| | LoRA | 55.60±0.28 | 10.51 | 6.23% | - | 140.86 |
| | QLoRA | 51.05±0.07 | 15.06 | 13.90% | 287.95 | 53.92 |
| | LoftQ | 51.20±0.13 | 14.91 | 13.65% | 286.16 | 53.35 |

for genome sequence classification problems, with input lengths ranging from 70 to 1000.

**Evaluation Metrics.** To evaluate the performance of outliers in our strategy, we report the *maximum infinity norm* $\|\mathbf{x}\|_\infty$ of the activation tensors $\mathbf{x}$ across all transformer layers as a metric for detecting outliers. Additionally, we present the *average kurtosis* of $\mathbf{x}$, calculated only from the output tensors from the Feed-Forward Network (FFN) layer and Layer Normalization. These two components are known to contain outliers, as confirmed by our experiments and prior studies (Hu et al., 2024a; Bondarenko et al., 2024; 2021). Both metrics have demonstrated a strong correlation with model quantizability (i.e., robustness to outliers) (Bondarenko et al., 2021; Chmiel et al., 2020). For pre-quantization performance, we also report the **FP16** (16-bit floating-point) Matthews correlation coefficient (MCC) score to assess model's downstream classification ability.

### 3.1 Post-Training Quantization (PTQ)

To assess the efficiency of our method for Post-Training Quantization (PTQ), we replace the standard attention layer in DNABERT-2 (Zhou et al., 2024) with the $\text{Softmax}_1$ activation function. We utilize the pre-trained checkpoints of these three models and fine-tune them at full rank following the procedure described in (Zhou et al., 2024). In this experiment, we evaluate the models on the test datasets using **FP16** precision and apply PTQ to measure performance degradation due to quantization. Each evaluation is

conducted three times with different random seeds, and we report the average and standard deviation for each metric.

**Baselines.** To evaluate the performance of our method against the official DNABERT-2 model, we also full fine-tune the official pretrained DNABERT-2 model [1] as a baseline on a downstream classification task to demonstrate the absolute performance of those three models. We further compare the performance of these three models on the same downstream classification task using **FP16** precision and four PTQ methods: Traditional W8A8 (Weights-8bit, Activations-8bit) as outlined in Bondarenko et al. (2024), SmoothQuant (Xiao et al., 2023), Outlier Suppression (Wei et al., 2022), and OmniQuant (Shao et al., 2024). With the exception of the W8A8 method, we evaluate and compare the quantization performance of SmoothQuant, Outlier Suppression, and OmniQuant at W8A8 (Weights-8bit, Activations-8bit) and W6A6 precision levels. Additionally, we present the quantization performance for W4A4 using OmniQuant and SmoothQuant. We use the same hyperparameters specified in their respective studies. This approach guarantees that our evaluations are conducted under standardized conditions, enabling precise comparisons and assessments of each quantization method.

**Results.** Referring to Table 1, it is clear that the GERM surpasses the DNABERT-2 in scenarios involving W4A4, W6A6 and W8A8 post-training quantization using state-of-the-art PTQ methods. Specifically, when both weights and activations are quantized to 8 bits (W8A8), GERM exhibits

a minimal average performance decline of only 4.82% on SmoothQuant. Additionally, the proposed strategy remains effective as the quantization bit size decreases. For example, when models are quantized to W4A4, GERM exhibits a minimal average performance decline of only 17.17% on OmniQuant, compared to an 94.78% drop in the DNABERT-2. This demonstrates that GERM outperforms the vanilla structure by enhancing the robustness of model quantization and improving performance across outlier-free methods like SmoothQuant and Outlier Suppression. Additionally, GERM-T exhibits strong quantization performance at both 8-bit and 6-bit levels, with a minimal performance drop, even smaller than that of GERM. The only exception is with W4A4 quantization, where GERM-T experiences a notable performance drop due to larger outliers in GERM-T compared to the GERM. The outlier metrics indicate the improvement in *average kurtosis* is minimal, though a substantial reduction in the *maximum infinity norm* compared to the DNABERT-2. Outlier metrics show that GERM reduces the average kurtosis by ∼92.14% and the maximum infinity norm by ∼82.77% across 27 datasets. Additionally, GERM-T achieves a reduction of approximately 7.20% in average kurtosis and 53.78% in the maximum infinity norm across the same datasets.

## 3.2 Low-Rank Adaptation

Fine-tuning models for downstream tasks is often computationally expensive. To enhance the efficiency of fine-tuning with fewer parameters, various parameter-efficient fine-tuning (PEFT) methods, such as Low-Rank Adaptation (LoRA), are commonly used. To assess the efficiency of our method for fine-tuning tasks with LoRA, we evaluate our framework on LoRA methods. We use the pretrained checkpoints of the three models and fine-tune them using multiple LoRA approaches following a similar procedure as described in Section 3.1. In this experiment, we evaluate the models on the test datasets using full-rank fine-tuning to compare the performance drop across various LoRA approaches. Each evaluation is conducted three times with different random seeds, and we report the average and standard deviation for each metric.

**LoRA Methods.** We compare our method with the vanilla version across three different LoRA methods: LoRA (Hu et al., 2022), QLoRA (Dettmers et al., 2024a), and LoftQ (Li et al., 2023). For the Full Fine-Tuning method, we fine-tune the model at full rank using mixed-precision FP16 training. For the LoRA method, following (Hu et al., 2022), we fine-tune the model with low-rank adaptations using a rank of 128 and an alpha value of 256. For the QLoRA and LoftQ methods, we fine-tune the model with quantized low-rank adaptations, maintaining the same rank and alpha values as in LoRA. Both QLoRA and LoftQ utilize 4-bit quantization

methods as described in (Dettmers et al., 2024a).

**Results.** In Table 2, our results highlight the effectiveness of GERM in low-rank adaptation. In most configurations, GERM significantly enhances fine-tuning performance. Specifically, GERM achieves an average performance improvement of **37.98%** in low-rank adaptation compared to DNABERT-2 model. Similarly, GERM-T shows an average performance improvement of **20.01%** over the same baseline. These results demonstrate that both GERM and GERM-T can greatly enhance low-rank adaptation for the model. When considering outlier metrics, we observe that LoRA exhibits significantly larger outlier values compared to full fine-tuning. Also, in most configurations, the outlier values in LoRA are much higher than those in QLoRA and LoftQ. One potential reason is that QLoRA and LoftQ utilize quantization to stabilize parameter updates and compress model representations, which helps minimize the amplification of extreme outlier values. Furthermore, GERM demonstrates a substantial reduction in outlier values compared to DNABERT-2, and GERM-T also experiences a significant decrease in the *maximum infinity norm*, though the reduction in *average kurtosis* remains limited.

Table 3: **Quantization Robustness Performance Comparison with Different Continual Learning Steps.** We evaluate the quantization robustness of different models, using SmoothQuant at 16-bit, 8-bit, and 6-bit quantization levels. The evaluation metric is the Matthews Correlation Coefficient (MCC) with the average performance drop following quantization also noted. The best results are highlighted in bold, and the second-best results are underlined.

| Method | #Bits | MCC (↑) | Avg Performance Drop (↓) |
|---|---|---|---|
| DNABERT-2 | 16W/16A | 59.11 | - |
| GERM | 16W/16A | 59.73 | - |
| Out20k | 16W/16A | 59.21 | - |
| GERM-T | 16W/16A | 59.30 | - |
| Out100k | 16W/16A | 60.56 | - |
| DNABERT-2 | 8W/8A | 36.51 | 38.23% |
| GERM | 8W/8A | 56.78 | 4.93% |
| Out20k | 8W/8A | 54.75 | 7.53% |
| GERM-T | 8W/8A | 57.52 | 3.00% |
| Out100k | 8W/8A | 58.77 | **2.96%** |
| DNABERT-2 | 6W/6A | 20.74 | 64.91% |
| GERM | 6W/6A | 56.48 | **5.44%** |
| Out20k | 6W/6A | 27.61 | 53.36% |
| GERM-T | 6W/6A | 28.32 | 52.24% |
| Out100k | 6W/6A | 30.44 | 49.74% |

## 3.3 Additional Experiments

In this section, we conduct additional experiments to evaluate the effectiveness of our method in various scenarios.

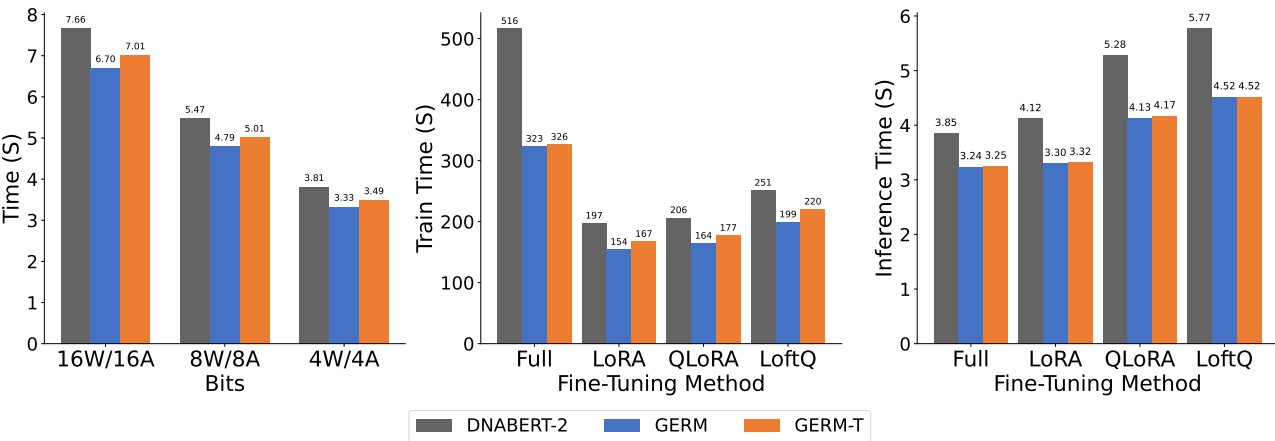

Figure 2: **Comparison of Performance in Resource-Constrained Computing Environments.** Comparison of three models on the quantization and fine-tuning task. All models were trained on the same computing infrastructure (Nvidia GeForce RTX 2080 Ti 11GB) for fair comparison. The training time represents the average time per epoch, with OmniQuant used as quantization example in this figure.

We conduct ablation studies to investigate the impact of different continual learning steps, the influence of adaptor rank. Also, we conduct a case study on model deployment and fine-tuning using a single 2080 Ti to evaluate model latency and fine-tuning speed per epoch.

**Impact of Different Continual Learning Steps.** To evaluate our proposed method's performance across various continual learning steps, we conduct experiments on three different step sizes: 20K, 40K, and 100K. We train the models for a total of 200K steps, employing a combination of vanilla and outlier-free pretraining as outlined in (Zhou et al., 2024) and (Hu et al., 2024a). To assess performance degradation after quantization using SmoothQuant, we perform full-rank fine-tuning of the models using the training datasets. Additionally, we assess the performance decline across models using three iterations of LoRA-based fine-tuning technology. We use the "Out**xxk**" prefix to indicate the number of continual learning steps applied to the outlier-free structure. For instance, Out100k represents a model trained with 100K continual learning steps using the outlier-free structure. This notation helps illustrate how different levels of continual learning impact the model's performance.In most configurations, GERM outperforms DNABERT-2 in both quantization robustness and low-rank adaptation. The only exception is in the 8-bit quantization scenario, where the Out100k and GERM-T models exhibit better performance and a smaller average performance drop than GERM. Overall, GERM-T achieves better near-optimal performance across all continual learning model settings. The results, as shown in Tables 3 and 4, demonstrate that our method outperforms the vanilla approach across all test sets. Also, we observe that GERM-T exhibits the most optimal performance drop during quantization and low-rank adaptation compared to other continual learning steps. The

larger performance drop observed in the model using 100K steps of continual learning, compared to GERM-T. It is attributed to the superior performance achieved through full fine-tuning. These results indicate that employing 40K continual learning steps in GERM-T is optimal for enhancing model performance in our approach.

**Performance of GERM on Alternative Transformer-based Models.** To assess the performance of our proposed method on alternative transformer-based GFMs, we conducted experiments on Nucleotide Transformer (NT) (Dalla-Torre et al., 2024), a significant genomic foundation model in this domain. The results, as shown in Tables 19 and 20, demonstrate that our method outperforms the vanilla approach across all test sets. GERM achieves an average performance improvement of 52.01% in low-rank adaptation and 67.69% in PTQ experiments. We also conduct experiments on the **NT-2.5B** model to demonstrate the scalability of GERM on larger-scale GFMs, as shown in Appendix D.7.

**Performance of GERM with Alternative Outlier Removal Methods.** To evaluate the performance of our proposed method against existing outlier removal techniques, we compare GERM with approaches such as clipped softmax and gated attention (Bondarenko et al., 2024). As shown in Appendix D.8, GERM achieves a performance improvement of 2.59% in 4-bit and 1.99% in 8-bit quantization.

**Performance of GERM with Different Adaptor Rank.** In our experiments, we evaluate the performance of our proposed method under the influence of adapter rank. All results are presented in Appendix D.1.

Table 4: **Low-Rank Adaptation Performance Comparison with Different Continual Learning Steps.** We evaluate the low-rank adaptation performance (LoRA, QLoRA, LoftQ) for various models with different continual learning steps. The evaluation metric is the Matthews Correlation Coefficient (MCC), and the average performance drop after adaptation is also shown. The best results are highlighted in bold, and the second-best results are underlined.

| Method | Fine-Tuning Method | MCC ($\uparrow$) | Avg Performance Drop ($\downarrow$) |
|---|---|---|---|
| DNABERT-2 | Full | 59.11 | - |
| GERM | Full | 59.73 | - |
| Out20k | Full | 59.21 | - |
| GERM-T | Full | 59.30 | - |
| Out100k | Full | 60.56 | - |
| DNABERT-2 | LoRA | 50.91 | 13.87% |
| GERM | LoRA | 56.78 | **4.94%** |
| Out20k | LoRA | 54.75 | 7.53% |
| GERM-T | LoRA | 55.60 | 6.24% |
| Out100k | LoRA | 56.61 | 6.52% |
| DNABERT-2 | QLoRA | 50.65 | 14.31% |
| GERM | QLoRA | 53.16 | **11.00%** |
| Out20k | QLoRA | 50.61 | 14.52% |
| GERM-T | QLoRA | 51.05 | 13.91% |
| Out100k | QLoRA | 51.24 | 15.39% |
| DNABERT-2 | LoftQ | 50.76 | 14.13% |
| GERM | LoftQ | 53.11 | **11.08%** |
| Out20k | LoftQ | 50.94 | 13.97% |
| GERM-T | LoftQ | 51.20 | 13.66% |
| Out100k | LoftQ | 50.77 | 16.17% |

### 3.4 Case Study: Performance in Resource-Constrained Computing Environments.

**Case Study 1: Performance in Single 2080-Ti GPU Computing Environments.** To demonstrate GERM's capability in resource-constrained environments, we conduct performance tests on a single NVIDIA GeForce RTX 2080 Ti 11GB GPU, where GERM requires only 5 minutes to fine-tune, demonstrating its practicality and efficiency. We compare GERM's performance with the DNABERT-2 model on the same downstream classification task using OmniQuant. Consistent hyperparameters from their respective studies are applied across all models. Additionally, we provide the per-epoch training time and inference time for the LoRA, QLoRA, and LoftQ fine-tuning methods. The results, as shown in Figure 2, show that both GERM and GERM-T achieve shorter full-rank fine-tuning times per epoch compared to DNABERT-2. Additionally, the model quantization latency for both GERM and GERM-T is lower than that of DNABERT-2, while delivering superior quantization performance. These observations indicate that GERM offers faster adaptation and requires fewer computational resources for

users during both the inference and fine-tuning processes.

**Case Study 2: Performance in CPU-Only Computing Environments.** To demonstrate GERM's capability in CPU-only computing environments, we perform performance tests on CPU-only devices. We compare GERM's per-epoch training and inference times for the LoRA and QLoRA fine-tuning methods. The results, presented in Appendix D.4, indicate that both GERM and GERM-T achieve shorter fine-tuning times per epoch compared to DNABERT-2, with the only exception being QLoRA when deployed, where the time is slightly longer.

## 4 Discussion and Conclusion

We introduce GERM, a versatile and accessible GFM designed to function on limited computational resources. By replacing the vanilla attention layer with an outlier-free layer, we eliminate outliers during both model pretraining and fine-tuning. This approach ensures robust quantization and enables effective low-rank adaptation. In addition to presenting a novel architecture that enhances DNABERT by mitigating outliers, GERM incorporates a compromise strategy with continual learning, eliminating the need for extensive retraining. Empirically, GERM achieves an average reduction of *average kurtosis* by ∼92.14% and the *maximum infinity norm* by ∼82.77% across 27 datasets. Additionally, GERM enhances model quantization robustness by decreasing the average quantization performance drop by 64.34% and the average low-rank adaptation performance drop by 37.98%. For the compromise model GERM-T, quantization robustness is improved by reducing the average quantization performance drop by 31.42% and the average low-rank adaptation performance drop by 20.01%.

**Limitations and Future Work.** While successful in many settings, our proposed GERM-T still faces challenges in eliminating outliers in GFM, leading to significant performance drops during 4-bit quantization and low-rank adaptation methods such as QLoRA and LoftQ. In future work, we aim to develop strategies that efficiently remove outliers without necessitating retraining from scratch. We provide a more detailed discussion of the limitations of GERM-T in Appendix E.

## Impact Statement

We believe this methodology presents an opportunity to strengthen the core of foundation models, including large language models, by improving robustness through quantization and enabling faster low-rank adaptation. However, this approach may also amplify biases in the training data, potentially leading to unfair or discriminatory outcomes for underrepresented groups.

## Acknowledgments

The authors would like to thank Yegna Jambunath for enlightening discussions on related topics, and Jiayi Wang for facilitating experimental deployments. The authors would like to thank the anonymous reviewers and program chairs for constructive comments.

Han Liu is partially supported by NIH R01LM1372201, NSF AST-2421845, Simons Foundation MPS-AI-00010513, AbbVie and Dolby. Haozheng Luo is partially supported by the OpenAI Researcher Access Program. This research was supported in part through the computational resources and staff contributions provided for the Quest high performance computing facility at Northwestern University which is jointly supported by the Office of the Provost, the Office for Research, and Northwestern University Information Technology. The content is solely the responsibility of the authors and does not necessarily represent the official views of the funding agencies.

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

# Supplementary Material

## A  Theoretical Analysis

In this section, we provide the expressive guarantee of Low-Rank Adaption for transformer model with $\text{Softmax}_1$. Moreover, we identify the conditions for the existence of low-rank adapters. We summarize our main findings in the following informal theorem.

> **Theorem A.1** (Informal). Let $f_0$ be the frozen model and $\bar{f}$ be the target model. Under certain non-singularity assumption and LoRA-rank conditions, their exist low-rank adapters such that the adopted model $f$ exactly equal to $\bar{f}$.

The formal version is stated in Theorem A.2. Then we provide a detailed illustration. We start with the definition of the target model $\bar{f}$ and the adopted model $f$.

> **Definition A.1** (Definition of target model $\bar{f}$ and adopted model $f$). For any input $X \in \mathbb{R}^{D \times N}$, where $D$ denotes the dimension of token embedding and $N$ denotes the number of tokens. We consider a $H$-heads transformers $\text{TF}_\theta$, consist of $L$-Transformer blocks and an output layer with parameter $\theta = \left( (W_{Ol}^h, W_{Vl}^h, W_{Kl}^h, W_{Ql}^h)_{h=1}^H, W_{1l}, W_{2l}, b_{1l}, b_{2l})_{l=1}^L, W_o \right)$. Specifically, we formulate it as
>
> $$\text{Hidden layer: Attn}(Z_{l-1}) = \sum_{h=1}^H \overline{W}_{Ol}^h \overline{W}_{Vl}^h \cdot \overline{Z}_{l-1} \cdot \text{Softmax}_1(\overline{Z}_{l-1}^\top \overline{W}_{Kl}^{h\top} \overline{W}_{Ql}^h \overline{Z}_{l-1}),$$
>
> $$Z_l = W_{2l} \cdot \text{ReLU}(W_{1l} \cdot \text{Attn}(Z_{l-1}) + b_{1l} \mathbf{1}_N^\top) + b_{2l} \mathbf{1}_N^\top$$
>
> $$\text{OutputLayer}: \text{TF}_\theta(X) = \text{Softmax}_1(W_o Z_L),$$
>
> where we define $Z_0 := X$. Here, $W_{1l}^h, W_{Vl}^h, W_{Kl}^h, W_{Ql}^h \in \mathbb{R}^{D \times D}$ are weight matrices in $l$-th attention layer. Further, $W_{1l}, W_{2l} \in \mathbb{R}^{D \times D}$ are weight matrices and $b_{1l}, b_{2l}$ are the bias vectors in the $l$-th feedforward layer. Then we define the

target model $\bar{f}$ and the adopted model $f$ are

$$\bar{f} := \mathrm{TF}_{\theta_T}, \quad \theta_T = \left( (\overline{W}_{Ol}^h, \overline{W}_{Vl}^h, \overline{W}_{Kl}^h, \overline{W}_{Ql}^h)_{h=1}^H, \overline{W}_{1l}, \overline{W}_{2l}, \bar{b}_{1l}, \bar{b}_{2l})_{l=1}^L, \overline{W}_o \right)$$

$$f := \mathrm{TF}_{\theta_A}, \quad \theta_A = ((W_{Ol}^h + \Delta W_{Ol}^h, W_{Vl}^h + \Delta W_{Vl}^h, W_{Kl}^h + \Delta W_{kl}^h, W_{Ql}^h + \Delta W_{Ql}^h)_{h=1}^H,$$
$$W_{1l} + \Delta W_{1l}, W_{2l} + \Delta W_{2l}, \widehat{b}_{1l}, \widehat{b}_{2l})_{l=1}^L, W_o + \Delta W_o).$$

Moreover, we define the best low-rank approximation for matrix $W$.

**Definition A.2** (Best Low-rank Approximation for $W$). For any matrix $W \in \mathbb{R}^{D \times D}$, the singular value decomposition (SVD) of $W$ is expressed as $W = UDV^\top$. Above $U, V \in \mathbb{R}^{D \times D}$ are orthonormal matrices and $D \in \mathbb{R}^{D \times D}$ is a diagonal matrix. Let the singular value of $W$ are denoted as $\sigma_1(W) \geq \ldots \geq \sigma_D(W) \geq 0$. When $d > D$, let $\sigma_d(W) = 0$. For any rank $r > 0$, we define

$$\mathrm{LRA}_r(W) := \sum_{i=1}^r \sigma_i(W) u_i v_i^\top,$$

where $u_i, v_i^\top$ are the $i$-th column of $U, V$, respectively.

According to Eckart & Young (1936) and Mirsky (1960), $\mathrm{LRA}_r(W)$ are the best rank-$r$ approximation in the Frobenius norm or the 2-norm of $W$. To present our results, we now introduce non-singularity assumption based on Definition A.2.

**Assumption A.1** (Non-Singularity). For a fixed $R \in [D]$, the weight matrices of both the target model, the frozen model and the following matrices are non-singular, for all $r \in [R]$. Specifically,

$$W_{Kl}^{h\top} W_{Ql}^h + \mathrm{LRA}_r(\overline{W}_{Kl}^{h\top} \overline{W}_{Ql}^h - W_{Kl}^{h\top} W_{Ql}^h), \text{ for all } h \in [H] \text{ and } l = 1,$$

$$W_{Kl}^{h\top} W_{Ql}^h + \mathrm{LRA}_r(W_{2,l-1}^{-1\top} \overline{W}_{2,l-1}^\top \overline{W}_{Kl}^{h\top} \overline{W}_{Ql}^h \overline{W}_{2,l-1} W_{2,l-1}^{-1} - W_{Kl}^{h\top} W_{Ql}^h), \text{ for all } h \in [H], l \in [L]\backslash\{1\},$$

$$W_{Ol}^h W_{Vl}^h + \mathrm{LRA}_r(W_{1l}^{-1} \overline{W}_{1l} \overline{W}_{Ol}^h \overline{W}_{Vl}^h - W_{Ol}^h W_{Vl}^h), \text{ for all } h \in [H] \text{ and } l = 1,$$

$$W_{Ol}^h W_{Vl}^h + \mathrm{LRA}_r(W_{1l}^{-1} \overline{W}_{1l} \overline{W}_{Ol}^h \overline{W}_{Vl}^h \overline{W}_{2,l-1} W_{2,l-1}^{-1} - W_{Ol}^h W_{Vl}^h), \text{ for all } h \in [H] \text{ and } l \in [L]\backslash\{1\},$$

$$W_o W_{2L} + \mathrm{LRA}_r(\overline{W}_o \overline{W}_{2L} - W_o W_{2L}),$$

are non-singular, where LRA denotes the rank-$r$ approximation follows Definition A.2.

Under a non-singularity assumption (Assumption A.1), we apply another helper lemma from Zeng & Lee (2024) to construct the weight matrices in Theorem A.2.

**Lemma A.1** (Exactly represent target model, Lemma 7 of Zeng & Lee (2024)). Define error matrix $E := \overline{W} - \prod_{l=1}^L W_l$, and denote its rank by $R_E = \mathrm{rank}(E)$. For a given LoRA-rank $R \in [D]$, assume that all the weight matrices of the frozen model $(W_l)_{l=1}^L$, and $\prod_{l=1}^L W_l + \mathrm{LRA}_r(E)$ are non-singular for all $r \leq R(L-1)$. Then, the approximation error

$$\min_{\Delta W_l : \mathrm{rank}(\Delta W_l) \leq R} \left\| \prod_{l=1}^L (W_l + \Delta W_l) - \overline{W} \right\|_2 = \sigma_{RL+1} \underbrace{\left( \overline{W} - \prod_{l=1}^L W_l \right)}_{\text{Error matrix } E}$$

and the optimal solution to the matrix approximation problem satisfies $\prod_{l=1}^L (W_l + \Delta W_l) = \prod_{l=1}^L W_l + \mathrm{LRA}_{RL \wedge R_E}(E)$. Therefore, when $R \geq \lceil \frac{R_E}{L} \rceil$, we have $\prod_{l=1}^L (W_l + \Delta W_l) = \overline{W}$, implying $f \equiv \bar{f}$.

With Assumption A.1 and Lemma A.1, we show that for any input $X \in \mathbb{R}^{D \times D}$, their exist a adapted model $f$ capable of approximating target model $\bar{f}$ exactly, i.e, $f(X) = \bar{f}(X)$.

**Theorem A.2** (Express capability of transformers). Suppose LoRA-rank $R \in [D]$. Let Assumption A.1 hold. Define the

rank-based functionality gap $G_i$ to i-th Outlier-Efficient Hopfield block ($i \in [L]$) or output layer ($i = L + 1$) as

$$
G_i = \begin{cases}
\max_h(\text{rank}(\overline{W}_{Ki}^{h\top}\overline{W}_{Qi}^h - W_{Ki}^{h\top}W_{Qi}^h)) \vee \max_h(\text{rank}(\overline{W}_{1i}\overline{W}_{Oi}^h\overline{W}_{Vi}^h - W_{1i}W_{Oi}^hW_{Vi}^h)), & i = 1, \\
\max_h(\text{rank}(\overline{W}_{2,i-1}^\top\overline{W}_{Ki}^{h\top}\overline{W}_{Qi}^h\overline{W}_{2,i-1} - W_{2,i-1}^\top W_{Ki}^{h\top}W_{Qi}^hW_{2,i-1}), \\
\quad \vee \max_h(\text{rank}(\overline{W}_{1i}\overline{W}_{Oi}^h\overline{W}_{Vi}^h\overline{W}_{2,i-1} - W_{1i}W_{Oi}^hW_{Vi}^hW_{2,i-1})), & 2 \leq i \leq L, \\
\text{rank}(\overline{W}_o\overline{W}_{2L} - W_oW_{2L}), & i = L + 1.
\end{cases}
$$

If $R \geq \max_{i \in [L+1]}\lceil\frac{G_i}{2}\rceil$, then there exists low-rank adapters with rank lower than $R$ $((\Delta W_{Kl}^h, \Delta W_{Ql}^h, \Delta W_{Vl}^h, \Delta W_{Ol}^h)_{h=1}^H)_{l=1}^L, \Delta W_{2L}, \Delta W_o$ with other low-rank adapters set to $O$, and updated bias vectors $(\widehat{b}_{1l}, \widehat{b}_{2l})_{l=1}^L$, such that for any input $X \in \mathbb{R}^{D \times N}$, the adapted model $f$ exactly approximates target model $\overline{f}$, i.e., $f(X) = \overline{f}(X)$.

*Proof.* We build our proof on Zeng & Lee (2024).

First, we ensure that, for each Outlier-Efficient Hopfield block, the output from the first feedforward layer in the target model matches that in the adapted model. Then, we select an appropriate output layer weight matrix to complete the proof.

We define $\overline{H}_l \in \mathbb{R}^{D \times N}$ and $\overline{Z}_l \in \mathbb{R}^{D \times N}$ as the intermediate and final outputs of the $l$-th transformer block in the target model $\overline{f}$, respectively. For any $l \in [L]$, they are formulated as

$$
\overline{H}_l := \text{ReLU}(\overline{W}_{1l}(\sum_{h=1}^H \overline{W}_{Ol}^h\overline{W}_{Vl}^h \cdot \overline{Z}_{l-1} \cdot \text{Softmax}_1(\overline{Z}_{l-1}^\top\overline{W}_{Kl}^{h\top}\overline{W}_{Ql}^h\overline{Z}_{l-1})) + \overline{b}_{1l}\mathbf{1}_N^\top),
$$
$$
\overline{Z}_l := \overline{W}_{2l}\overline{H}_l + \overline{b}_{2l}\mathbf{1}_N^\top.
$$

Correspondingly, we introduce $\widehat{H}_l$ and $\widehat{Z}_l$ to denote the intermediate output of the first feedforward layer and the final output of the $l$-th Outlier-Efficient Hopfield block for the adapted model $f$,

$$
\widehat{H}_l = \text{ReLU}(W_{1l}(\sum_{h=1}^H (W_{Ol}^h + \Delta W_{Ol}^h)(W_{Vl}^h + \Delta W_{Vl}^h) \cdot \widehat{Z}_{l-1} \tag{A.1}
$$
$$
\cdot \text{Softmax}_1(\widehat{Z}_{l-1}^\top(W_{Kl}^h + \Delta W_{Kl}^h)^\top(W_{Ql}^h + \Delta W_{Ql}^h)\widehat{Z}_{l1}) + \widehat{b}_{1l}\mathbf{1}_N^\top),
$$
$$
\widehat{Z}_l = W_{2l}\widehat{H}_l + \widehat{b}_{2l}\mathbf{1}_N^\top, \tag{A.2}
$$

for any $l \in [L]$. Note that $\overline{Z}_0 = \widehat{Z}_0 = X$. Next, we inductively construct the adapter weight matrices $((\Delta W_{Ol}^h, \Delta W_{Vl}^h, \Delta W_{Kl}^h, \Delta W_{Ql}^h)_{h=1}^H, \widehat{b}_{1l}, \widehat{b}_{2l})_{l=1}^L$ such that $\widehat{H}_l = \overline{H}_l$ for all $l \in [L]$. We then select the low-rank adapters for $W_{2L}$ and the $W_o$ to approximate the output of the target model. For unmentioned low-rank adapters, we set them as $O$.

**When $l = 1$.** To achieve $\widehat{H}_l = \overline{H}_l$ for all $X$, the following conditions must be satisfied:

- Bias Vector: $\widehat{b}_{1l} = \overline{b}_{1l}$,

- Query and Key: $(W_{Kl}^h + \Delta W_{Kl}^h)^\top(W_{Ql}^h + \Delta W_{Ql}^h) = \overline{W}_{Kl}^{h\top}\overline{W}_{Ql}^h$,

- Value and First Feedforward Layer: $(W_{Ol}^h + \Delta W_{Ol}^h)(W_{Vl}^h + \Delta W_{Vl}^h) = W_{1l}^{-1}\overline{W}_{1l}\overline{W}_{Ol}^h\overline{W}_{Vl}^h$,

It is simple to check that we only need to set $\widehat{b}_{1l} = \overline{b}_{1l}$ to, and select rank-$R$ or lower matrices $\Delta W_{Kl}^h, \Delta W_{Ql}^h, \Delta W_{Ol}^h, \Delta W_{Vl}^h$ as suggested by Lemma A.1. This ensures $\widehat{H}_l = \overline{H}_l$ for $l = 1$.

**When $l > 1$.** For the cases $l = 2, \ldots, L$, we assume the induction hypothesis holds for $l - 1$, which is $\widehat{H}_{l-1} = \overline{H}_{l-1}$. We let $\widehat{b}_{2,l-1} = W_{2,l-1}\overline{W}_{2,l-1}^{-1}\overline{b}_{2,l-1}$, then it holds,

$$
\widehat{Z}_{l-1} = W_{2,l-1}\overline{W}_{2,l-1}^{-1}\overline{Z}_{l-1}. \tag{A.3}
$$

Substituting (A.3) into (A.1) and (A.2), the necessary conditions become:

- Bias Vector: $\widehat{b}_{1l} = \overline{b}_{1l}$,

- Query and Key: $(W_{Kl}^h + \Delta W_{Kl}^h)^\top (W_{Ql}^h + \Delta W_{Ql}^h) = W_{2,l-1}^{-1\top} \overline{W}_{2,l-1}^\top \overline{W}_{Kl}^{h\top} \overline{W}_{Ql}^h \overline{W}_{2,l-1} W_{2,l-1}^{-1}$,

- Value and Output Projection: $(W_{Ol}^h + \Delta W_{Ol}^h)(W_{Vl}^h + \Delta W_{Vl}^h) = W_{1l}^{-1} \overline{W}_{1l} \overline{W}_{Ol}^h \overline{W}_{Vl}^h \overline{W}_{2,l-1} W_{2,l-1}^{-1}$.

By setting $\widehat{b}_{1l} = \overline{b}_{1l}$ and adjusting $\Delta W_{Kl}^h, \Delta W_{Ql}^h, \Delta W_{Ol}^h, \Delta W_{Vl}^h$ for all $h \in [H]$ based on Lemma A.1, we satisfy all three conditions above, thereby obtaining $\widehat{H}_l = \overline{H}_l$ for $l \in [L] \backslash \{1\}$.

**Output Layer Analysis.** By applying the induction method, we have established $\widehat{H}_l = \overline{H}_l$ for all $l \in [L]$. We only need to select appropriate weight matrices to ensure that $\overline{f}(X) = f(X)$ for all $X \in \mathcal{X}$. The final output of the target model $\overline{f}$ with input $X$ can be written as

$$\overline{f}(X) = \text{Softmax}_1(\overline{W}_o \overline{Z}_L) = \text{Softmax}_1(\overline{W}_o(\overline{W}_{2L} \overline{H}_L + \overline{b}_{2L} \mathbf{1}_N^\top)).$$

Similarly, the final output of the adapted model $f$ with input $X$ can be written as

$$\begin{aligned} f(X) &= \text{Softmax}_1((W_o + \Delta W_o)\widehat{Z}_L) \\ &= \text{Softmax}_1((W_o + \Delta W_o)((W_{2L} \Delta W_{2L})\widehat{H}_L + \widehat{b}_{2L} \mathbf{1}_N^\top)). \end{aligned}$$

To achieve $\overline{f}(X) = f(X)$, we select $\Delta W_{2L}$ and $\Delta W_o$ based on Lemma A.1, and let $\widehat{b}_{2L} = (W_o + \Delta W_o)^{-1} \overline{W}_o \overline{b}_{2L}$, where $W_o + \Delta W_o$ is invertible as shown in the proof of Lemma A.1. Combining above, we complete the proof. □

# B    Experimental Setup

## B.1    Computational Resource

We perform all experiments using 2 NVIDIA A100 GPU with 80GB of memory and a 24-core Intel(R) Xeon(R) Gold 6338 CPU operating at 2.00GHz. Our code is developed in PyTorch and utilizes the Hugging Face Transformer Library for experimental execution.

## B.2    Hyperparameters

We present the hyperparameters used in the fine-tuning stage for each model. We use **AdamW** (Loshchilov & Hutter, 2019) as the optimizer. Most of the other hyperparameters remain the same across all models and datasets, including a batch size of 32, a warmup step of 50, and a weight decay of 0.01. A learning rate of $3e^{-5}$ is used for all models during fine-tuning. For low-rank adaptation, we use a learning rate of $1e^{-4}$, with a LoRA rank of 8 and LoRA alpha set to 16. For each task, we use different training steps as shown in Table 5. During pre-training, the model is trained for 200,000 steps with a batch size of 1024 and a maximum sequence length of 512, using the AdamW optimizer with $\beta_1 = 0.9$, $\beta_2 = 0.98$, and $\epsilon = 1e^{-6}$. The pre-training stage takes approximately 4 days using 2 NVIDIA A100 80G GPUs.

Table 5: **The number of training steps.** We present the number of training steps we use in our experiments. In the task of Transcription Factor Prediction on the Mouse genome, we train the model for 1000 steps on each dataset.

|        | EMP | TF-M | CVC | TF-H | PD-tata | PD-o | CPD-tata | CPD-o | SSP |
|--------|-----|------|-----|------|---------|------|----------|-------|-----|
| Epochs | 3   | 1k   | 8   | 3    | 10      | 4    | 10       | 4     | 5   |

## B.3    Downstream Tasks Across Different Models

We analyze the downstream tasks of various genomic foundation models (GFMs), specifically comparing DNABERT-2 (Zhou et al., 2024), HyenaDNA (Nguyen et al., 2024b), and Nucleotide Transformer (Dalla-Torre et al., 2024). As shown in Table 6, all of these GFMs utilize classification tasks as their primary downstream applications. Additionally, we analyze related GenBench datasets (Liu et al., 2025) and find that, uniquely, GenBench includes some regression downstream tasks, providing a broader evaluation spectrum.

Table 6: **Comparison of Models (Benchmarks) and Their Tasks.**

| Model | Tasks | Classification-Only |
|---|---|---|
| DNABERT-2 | GUE (28 Classification tasks) | Yes |
| Nucleotide Transformer | Nucleotide Transformer Benchmark (18 Classification tasks) | Yes |
| HyenaDNA | GenBench (Classification-Only) + Nucleotide Transformer Benchmark | Yes |
| GenBench | Classification + Regression (e.g., Drosophila Enhancer Activity Prediction) | No |

## C Formula of Average Kurtosis

As shown in Bondarenko et al. (2024); Hu et al. (2024a), average kurtosis is a significant metric for measuring outliers. Kurtosis is a statistical measure that quantifies the "tailedness" of a distribution relative to a normal distribution. The formula for kurtosis is:

$$K = \frac{n \sum_{i=1}^{n}(x_i - \bar{x})^4}{\left(\sum_{i=1}^{n}(x_i - \bar{x})^2\right)^2},$$

where $x_i$ represents the data points, $\bar{x}$ is the mean, and $n$ is the number of data points. As a result, when a distribution has higher kurtosis, it indicates that the data distribution has heavier tails. In other words, there are more outlier values present in the distribution. The GERM reduces kurtosis by evenly distributing attention across informative tokens, rather than focusing excessively on specific outlier tokens (e.g., delimiters or eos markers).

## D Additional Numerical Experiments

### D.1 Influence of Adaptor Rank

We perform an in-depth analysis to assess the performance of our proposed method across different ranks when implementing Low-rank Adaptation (LoRA), comparing it to the standard vanilla approach. We compare the model performance with different rank value of LoRA and keep the alpha value double of the rank value. The results, as shown in Table 7, demonstrate that our method outperforms the vanilla approach across all tested ranks. We observe that a rank of 8 provides optimal performance compared to the vanilla method. The lack of further performance improvements with higher ranks is because a higher rank in LoRA usually introduces more trainable parameters into the model. This leads to a significant performance improvement over the vanilla structure, thereby narrowing the performance gap with GERM.

Table 7: **Comparison of Different Ranks Using LoRA.** We conduct experiments to evaluate how different ranks affect the performance of LoRA. The evaluation metric used is Matthews Correlation Coefficient (MCC). We measure the average performance decline following low-rank adaptation, with best results highlighted in bold.

| Method | Fine-Tuning Method | Rank | MCC |
|---|---|---|---|
| DNABERT-2 | Full | N/A | 59.11 |
| GERM | Full | N/A | **59.73** |
| GERM-T | Full | N/A | 59.30 |
| DNABERT-2 | LoRA | 16 | 55.71 |
| GERM | LoRA | 16 | **58.91** |
| GERM-T | LoRA | 16 | 57.41 |
| DNABERT-2 | LoRA | 8 | 52.87 |
| GERM | LoRA | 8 | **57.27** |
| GERM-T | LoRA | 8 | 55.60 |
| DNABERT-2 | LoRA | 4 | 51.02 |
| GERM | LoRA | 4 | **55.64** |
| GERM-T | LoRA | 4 | 52.07 |

## D.2 All Results in Low-rank Adaptation

In this section, we present a comprehensive evaluation of various models under different low-rank adaptation (LoRA) strategies. The experiments compare DNABERT-2, GERM-T, and GERM models across multiple biological prediction tasks, including epigenetic marks prediction, promoter detection, and transcription factor prediction in both human and mouse datasets. The adaptation methods assessed include Full Fine-tuning, LoRA, QLoRA, and LoftQ.

Table 8: **Performance Comparison of Full Fine-tuning with DNABERT-2.** This table shows the performance of all models on the full fine-tuning task.

| Model | Epigenetic Marks Prediction | | | | | |
|---|---|---|---|---|---|---|
| | H3 | H3K14ac | H3K36me3 | H3K4me1 | H3K4me2 | H3K4me3 |
| DNABERT-2 | 75.03 | 33.07 | 48.63 | 33.67 | 31.63 | **24.31** |
| GERM-T | 75.02 | 47.31 | 51.32 | **34.53** | 27.72 | 23.19 |
| GERM | **75.58** | **50.36** | **53.13** | 33.36 | **36.02** | 23.97 |

| Model | Epigenetic Marks Prediction | | | | Promoter Detection | | |
|---|---|---|---|---|---|---|---|
| | H3K79me3 | H3K9ac | H4 | H4ac | all | notata | tata |
| DNABERT-2 | **60.88** | 49.87 | 77.67 | 26.87 | **82.45** | 90.29 | 58.20 |
| GERM-T | 59.61 | 52.63 | 80.20 | 45.37 | 80.82 | 90.24 | 59.27 |
| GERM | 59.80 | **55.08** | **81.25** | **48.36** | 79.40 | **90.51** | **59.75** |

| Model | Transcription Factor Prediction (Human) | | | | | Core Promoter Detection | | |
|---|---|---|---|---|---|---|---|---|
| | 0 | 1 | 2 | 3 | 4 | all | notata | tata |
| DNABERT-2 | 66.51 | 70.10 | 57.43 | 39.67 | 71.35 | **66.61** | 65.34 | 71.98 |
| GERM-T | **68.17** | 68.45 | **62.85** | 50.62 | **72.25** | 48.96 | 65.09 | **75.90** |
| GERM | 67.29 | **70.88** | 58.21 | **51.75** | 72.21 | 51.91 | 63.07 | 70.40 |

| Model | Transcription Factor Prediction (Mouse) | | | | | Virus | Splice |
|---|---|---|---|---|---|---|---|
| | 0 | 1 | 2 | 3 | 4 | Covid | Reconstruct |
| DNABERT-2 | **45.61** | **80.20** | **78.05** | 72.70 | **41.94** | 66.17 | 68.85 |
| GERM-T | 40.90 | 57.18 | 71.25 | 70.25 | 36.88 | **66.69** | **77.60** |
| GERM | 41.98 | 59.19 | 65.53 | 72.58 | 38.68 | 66.02 | 76.59 |

Table 9: **Performance Comparison of LoRA with DNABERT-2.** This table shows the performance of all models on the low-rank adaptation (LoRA) task.

| | Epigenetic Marks Prediction | | | | | |
| Model | H3 | H3K14ac | H3K36me3 | H3K4me1 | H3K4me2 | H3K4me3 |
|---|---|---|---|---|---|---|
| DNABERT-2 | 65.93 | 35.97 | 44.19 | 38.93 | 28.30 | 21.86 |
| GERM-T | 69.98 | **42.59** | 48.34 | **41.15** | 29.11 | 24.45 |
| GERM | **70.61** | 37.12 | **48.57** | 35.46 | **30.20** | **25.62** |

| | Epigenetic Marks Prediction | | | | Promoter Detection | | |
| Model | H3K79me3 | H3K9ac | H4 | H4ac | all | notata | tata |
|---|---|---|---|---|---|---|---|
| DNABERT-2 | 55.69 | 45.65 | 74.15 | 33.52 | 81.84 | 89.72 | 53.64 |
| GERM-T | **58.25** | **49.96** | 78.08 | **36.18** | **84.19** | **91.71** | **60.18** |
| GERM | 57.99 | 47.54 | **78.37** | 35.55 | 82.97 | 90.35 | 58.51 |

| | Transcription Factor Prediction (Human) | | | | | Core Promoter Detection | | |
| Model | 0 | 1 | 2 | 3 | 4 | all | notata | tata |
|---|---|---|---|---|---|---|---|---|
| DNABERT-2 | 61.68 | 69.70 | 38.80 | 42.17 | 58.94 | **66.61** | **65.34** | 65.34 |
| GERM-T | 65.21 | 69.47 | **57.42** | 46.86 | 66.18 | 48.96 | 65.09 | **75.90** |
| GERM | **66.77** | **71.03** | 55.95 | **48.65** | **69.63** | 51.91 | 63.07 | 70.40 |

| | Transcription Factor Prediction (Mouse) | | | | | Virus | Splice |
| Model | 0 | 1 | 2 | 3 | 4 | Covid | Reconstruct |
|---|---|---|---|---|---|---|---|
| DNABERT-2 | **45.61** | **80.20** | **78.05** | 72.70 | **41.94** | 13.15 | 56.16 |
| GERM-T | 40.90 | 57.18 | 71.25 | 70.25 | 36.88 | 23.36 | 60.85 |
| GERM | 41.98 | 59.19 | 65.53 | 72.58 | 38.68 | **41.37** | **64.62** |

Table 10: **Performance Comparison of QLoRA with DNABERT-2.** This table shows the performance of all models on the QLoRa task.

| Model | Epigenetic Marks Prediction | | | | | |
| | H3 | H3K14ac | H3K36me3 | H3K4me1 | H3K4me2 | H3K4me3 |
| --- | --- | --- | --- | --- | --- | --- |
| DNABERT-2 | 57.67 | 32.22 | 38.84 | 29.17 | **28.74** | 20.47 |
| GERM-T | 59.52 | 33.03 | 35.84 | 28.85 | 28.02 | **21.53** |
| GERM | **60.55** | **37.30** | **39.52** | **33.22** | 26.61 | 21.29 |

| Model | Epigenetic Marks Prediction | | | | Promoter Detection | | |
| | H3K79me3 | H3K9ac | H4 | H4ac | all | notata | tata |
| --- | --- | --- | --- | --- | --- | --- | --- |
| DNABERT-2 | 53.11 | 41.69 | 67.02 | 28.92 | 81.27 | 88.66 | 54.98 |
| GERM-T | 54.22 | 44.76 | 72.82 | 29.68 | 81.15 | 87.58 | 55.07 |
| GERM | **56.80** | **47.41** | **74.76** | **30.85** | **82.13** | **90.02** | **56.97** |

| Model | Transcription Factor Prediction (Human) | | | | | Core Promoter Detection | | |
| | 0 | 1 | 2 | 3 | 4 | all | notata | tata |
| --- | --- | --- | --- | --- | --- | --- | --- | --- |
| DNABERT-2 | **62.62** | 66.97 | 49.46 | 41.86 | 63.86 | 60.41 | 64.26 | **63.24** |
| GERM-T | 62.00 | 67.31 | 50.84 | 41.57 | **65.87** | 60.48 | **65.14** | 55.80 |
| GERM | 62.10 | **69.00** | **53.28** | **44.80** | 65.71 | **61.15** | 64.73 | 57.27 |

| Model | Transcription Factor Prediction (Mouse) | | | | | Virus | Splice |
| | 0 | 1 | 2 | 3 | 4 | Covid | Reconstruct |
| --- | --- | --- | --- | --- | --- | --- | --- |
| DNABERT-2 | 36.56 | 75.32 | 70.55 | 50.47 | **35.41** | 6.44 | 49.99 |
| GERM-T | 35.63 | 72.05 | 70.82 | 48.77 | 34.46 | 14.81 | 51.87 |
| GERM | **37.01** | **76.59** | **72.95** | **52.34** | 33.48 | **16.72** | **59.12** |

Across all adaptation methods, GERM consistently outperforms DNABERT-2 and GERM-T in various prediction tasks, particularly excelling in epigenetic marks prediction and promoter detection. This consistent superiority suggests that GERM possesses a higher degree of flexibility and effectiveness in low-rank adaptation scenarios. While DNABERT-2 and GERM-T show competitive performance in certain tasks and adaptation methods, GERM's robust performance across diverse biological datasets and adaptation strategies underscores its potential as a more reliable tool for complex genomic predictions.

Table 11: **Performance Comparison of LoftQ with DNABERT-2.** This table shows the performance of all models on the LoftQ task.

| Model | Epigenetic Marks Prediction | | | | | |
| | H3 | H3K14ac | H3K36me3 | H3K4me1 | H3K4me2 | H3K4me3 |
| --- | --- | --- | --- | --- | --- | --- |
| DNABERT-2 | 58.92 | 32.36 | 36.59 | 28.84 | 27.19 | 18.46 |
| GERM-T | 60.03 | 32.03 | 37.23 | 28.78 | **27.94** | 20.16 |
| GERM | **62.17** | **36.79** | **39.45** | **33.12** | 27.37 | **21.84** |

| Model | Epigenetic Marks Prediction | | | | Promoter Detection | | |
| | H3K79me3 | H3K9ac | H4 | H4ac | all | notata | tata |
| --- | --- | --- | --- | --- | --- | --- | --- |
| DNABERT-2 | **60.88** | 46.21 | 67.86 | 29.63 | 80.44 | 88.71 | 56.79 |
| GERM-T | 59.61 | 52.63 | 80.20 | 30.57 | 80.10 | 88.78 | 58.52 |
| GERM | 59.80 | **55.08** | **81.25** | **31.23** | **82.00** | **89.81** | **59.86** |

| Model | Transcription Factor Prediction (Human) | | | | | Core Promoter Detection | | |
| | 0 | 1 | 2 | 3 | 4 | all | notata | tata |
| --- | --- | --- | --- | --- | --- | --- | --- | --- |
| DNABERT-2 | 63.30 | 67.63 | 50.39 | 43.41 | 64.09 | 59.80 | 64.24 | 56.93 |
| GERM-T | **63.35** | 66.70 | 51.82 | 40.13 | 64.13 | 60.74 | 64.50 | 57.46 |
| GERM | 62.08 | **68.84** | **54.25** | **44.70** | **64.57** | **61.43** | **65.12** | **59.10** |

| Model | Transcription Factor Prediction (Mouse) | | | | | Virus | Reconstruct |
| | 0 | 1 | 2 | 3 | 4 | Covid | Reconstruct |
| --- | --- | --- | --- | --- | --- | --- | --- |
| DNABERT-2 | 35.61 | 74.25 | 71.16 | 51.63 | 34.37 | 8.31 | 52.17 |
| GERM-T | 39.46 | 75.27 | 71.24 | **57.41** | 34.28 | 5.01 | 55.89 |
| GERM | **42.13** | **76.55** | **71.96** | 51.46 | **36.04** | **9.14** | **60.12** |

## D.3 All Results in Post-Training Quantization

In this section, we present the results of all experiments conducted for the post-training quantization (PTQ).

Table 12: **Performance Comparison of Outlier Suppression with DNABERT-2.** This table shows the performance of all models on the Outlier Suppression.

| Model | Bits | Epigenetic Marks Prediction | | | | | |
|---|---|---|---|---|---|---|---|
| | | H3 | H3K14ac | H3K36me3 | H3K4me1 | H3K4me2 | H3K4me3 |
| DNABERT-2 | 8W/8A | 31.81 | 0.11 | 16.17 | 0.74 | 14.33 | 1.69 |
| | 6W/6A | 26.56 | 1.44 | 13.76 | 2.12 | 13.50 | 3.04 |
| GERM-T | 8W/8A | 65.55 | 33.07 | 34.81 | 6.82 | 23.29 | 15.60 |
| | 6W/6A | 62.50 | 11.13 | 38.51 | 27.66 | 21.31 | 15.68 |
| GERM | 8W/8A | 66.29 | 27.78 | 28.54 | 32.06 | 13.47 | 14.77 |
| | 6W/6A | 60.19 | 13.89 | 21.67 | 28.26 | 3.72 | 5.49 |

| Model | Bits | Epigenetic Marks Prediction | | | | Promoter Detection | | |
|---|---|---|---|---|---|---|---|---|
| | | H3K79me3 | H3K9ac | H4 | H4ac | all | notata | tata |
| DNABERT-2 | 8W/8A | 17.90 | 13.30 | 2.30 | 11.46 | 29.46 | 21.55 | 26.88 |
| | 6W/6A | 12.66 | 14.30 | 2.30 | 10.68 | 25.19 | 18.33 | 23.10 |
| GERM-T | 8W/8A | 46.91 | 32.02 | 75.01 | 32.99 | 53.15 | 46.88 | 29.94 |
| | 6W/6A | 44.79 | 15.91 | 76.85 | 11.45 | 71.60 | 82.13 | 44.51 |
| GERM | 8W/8A | 44.44 | 29.83 | 77.00 | 19.42 | 75.36 | 83.04 | 31.92 |
| | 6W/6A | 22.82 | 11.73 | 21.67 | 3.77 | 69.04 | 82.75 | 23.43 |

| Model | Bits | Transcription Factor Prediction (Human) | | | | | Core Promoter Detection | | |
|---|---|---|---|---|---|---|---|---|---|
| | | tf0 | tf1 | tf2 | tf3 | tf4 | all | notata | tata |
| DNABERT-2 | 8W/8A | 41.72 | 41.84 | 40.79 | 6.84 | 42.45 | 54.50 | 28.58 | 54.53 |
| | 6W/6A | 41.43 | 42.05 | 39.64 | 9.96 | 43.17 | 55.54 | 26.75 | 56.20 |
| GERM-T | 8W/8A | 35.24 | 71.82 | 64.96 | 37.81 | 23.54 | 58.13 | 60.69 | 57.54 |
| | 6W/6A | 54.18 | 76.65 | 67.42 | 27.09 | 23.54 | 57.69 | 56.73 | 73.47 |
| GERM | 8W/8A | 54.37 | 52.53 | 41.99 | 45.58 | 59.72 | 47.09 | 55.61 | 66.39 |
| | 6W/6A | 54.62 | 51.94 | 41.34 | 44.74 | 60.88 | 46.63 | 54.85 | 66.47 |

| Model | Bits | Transcription Factor Prediction (Mouse) | | | | | Virus | Splice |
|---|---|---|---|---|---|---|---|---|
| | | 0 | 1 | 2 | 3 | 4 | Covid | Reconstruct |
| DNABERT-2 | 8W/8A | 35.39 | 23.62 | 52.89 | 41.51 | 23.73 | 23.74 | 6.69 |
| | 6W/6A | 34.55 | 23.20 | 55.55 | 39.24 | 25.13 | 25.55 | 6.87 |
| GERM-T | 8W/8A | 35.24 | 71.82 | 64.96 | 37.81 | 23.54 | 19.42 | 30.25 |
| | 6W/6A | 54.18 | 76.65 | 67.42 | 27.09 | 29.01 | 53.21 | 28.47 |
| GERM | 8W/8A | 50.97 | 48.73 | 41.80 | 50.21 | 30.48 | 61.04 | 36.76 |
| | 6W/6A | 49.96 | 48.50 | 39.87 | 47.53 | 28.95 | 59.94 | 36.79 |

Table 13: **Performance Comparison of SmoothQuant with DNABERT-2.** This table shows the performance of all models on the SmoothQuant.

| Model | Bits | Epigenetic Marks Prediction | | | | | |
|---|---|---|---|---|---|---|---|
| | | H3 | H3K14ac | H3K36me3 | H3K4me1 | H3K4me2 | H3K4me3 |
| DNABERT-2 | 8W/8A | 44.40 | -1.47 | 42.17 | 13.12 | 23.30 | 8.39 |
| | 6W/6A | 28.07 | -0.08 | 22.77 | 2.96 | 15.73 | 0.10 |
| | 4W/4A | -1.85 | 0.83 | -1.09 | -0.94 | **2.31** | -0.90 |
| GERM-T | 8W/8A | **72.44** | 46.34 | 50.15 | 28.18 | 26.59 | 23.25 |
| | 6W/6A | 27.19 | 8.68 | 28.30 | 2.15 | 14.21 | 4.82 |
| | 4W/4A | 0.00 | -2.43 | 0.94 | 1.03 | 0.80 | -0.07 |
| GERM | 8W/8A | 72.07 | **49.89** | **52.95** | **33.11** | **32.28** | **24.87** |
| | 6W/6A | **72.23** | **48.60** | **53.67** | **31.94** | **32.09** | **24.76** |
| | 4W/4A | **18.17** | **6.83** | **20.84** | **4.03** | 0.00 | **1.43** |

| Model | Bits | Epigenetic Marks Prediction | | | | Promoter Detection | | |
|---|---|---|---|---|---|---|---|---|
| | | H3K79me3 | H3K9ac | H4 | H4ac | all | notata | tata |
| DNABERT-2 | 8W/8A | 55.54 | 14.16 | 2.30 | 10.19 | 44.80 | 58.87 | 38.19 |
| | 6W/6A | 25.14 | 20.25 | 3.99 | 0.21 | 78.44 | 57.23 | 0.54 |
| | 4W/4A | 0.45 | -0.90 | -5.07 | 1.20 | 0.00 | -1.78 | -5.42 |
| GERM-T | 8W/8A | 59.78 | 48.39 | 76.85 | 43.62 | 73.81 | 81.40 | 58.84 |
| | 6W/6A | 49.94 | 41.77 | 40.94 | 20.73 | 72.97 | 70.85 | 12.03 |
| | 4W/4A | -2.53 | 2.26 | 0.00 | **3.10** | -1.31 | -1.16 | 0.00 |
| GERM | 8W/8A | **61.36** | **50.31** | **79.61** | **48.66** | **80.60** | **92.51** | 58.08 |
| | 6W/6A | **62.15** | **49.05** | **78.99** | **48.37** | 79.95 | **92.18** | **58.73** |
| | 4W/4A | **21.16** | **0.00** | **56.10** | 0.17 | **35.32** | **76.43** | **8.04** |

| Model | Bits | Transcription Factor Prediction (Human) | | | | | Core Promoter Detection | | |
|---|---|---|---|---|---|---|---|---|---|
| | | 0 | 1 | 2 | 3 | 4 | all | notata | tata |
| DNABERT-2 | 8W/8A | 45.06 | 48.20 | 47.29 | 4.70 | 52.27 | 60.82 | 41.31 | 68.15 |
| | 6W/6A | 28.41 | 29.85 | 34.60 | 6.51 | 20.96 | 47.72 | 36.38 | 31.69 |
| | 4W/4A | 5.55 | -2.74 | -1.51 | -0.72 | 4.52 | 1.53 | -3.00 | -0.15 |
| GERM-T | 8W/8A | 57.74 | 52.51 | **53.17** | **46.36** | 65.22 | **64.15** | 61.01 | **75.31** |
| | 6W/6A | 46.18 | 43.36 | 38.43 | 36.12 | 43.69 | **52.24** | 60.68 | 28.46 |
| | 4W/4A | 2.33 | -3.19 | -2.42 | 1.79 | 4.53 | 0.78 | **3.33** | -5.73 |
| GERM | 8W/8A | **58.54** | **52.44** | 48.14 | 45.42 | **66.42** | 49.26 | **64.46** | 70.31 |
| | 6W/6A | **57.55** | **51.81** | 48.29 | 45.19 | **65.87** | 47.55 | **64.35** | 69.67 |
| | 4W/4A | **43.41** | **35.76** | **25.56** | **9.77** | **22.75** | **21.99** | 40.51 | **12.46** |

| Model | Bits | Transcription Factor Prediction (Mouse) | | | | | Virus | Splice |
|---|---|---|---|---|---|---|---|---|
| | | 0 | 1 | 2 | 3 | 4 | Covid | Reconstruct |
| DNABERT-2 | 8W/8A | 45.74 | 47.95 | 67.98 | 57.32 | 29.55 | 27.30 | 24.70 |
| | 6W/6A | 13.28 | 25.38 | 30.48 | 7.67 | 12.96 | -0.44 | 0.00 |
| | 4W/4A | 0.06 | -1.95 | -15.92 | -2.76 | 2.59 | 0.13 | 0.95 |
| GERM-T | 8W/8A | 46.21 | **82.36** | **75.48** | **67.31** | 33.53 | **66.74** | 73.63 |
| | 6W/6A | 0.00 | 27.14 | 42.68 | 21.42 | 6.52 | 7.95 | 0.00 |
| | 4W/4A | -1.22 | 1.22 | 0.00 | 0.00 | 1.49 | -0.22 | 0.00 |
| GERM | 8W/8A | **46.87** | 53.88 | 54.71 | 65.01 | **38.48** | 64.51 | **75.14** |
| | 6W/6A | **44.96** | 60.08 | 53.97 | 64.14 | 37.28 | 64.69 | 73.20 |
| | 4W/4A | **8.07** | **26.40** | **10.00** | **27.02** | **9.80** | **19.48** | 0.00 |

Table 14: **Performance Comparison of OmniQuant with DNABERT-2.** This table shows the performance of all models on the OmniQuant.

| Model | Bits | Epigenetic Marks Prediction | | | | | |
| | | H3 | H3K14ac | H3K36me3 | H3K4me1 | H3K4me2 | H3K4me3 |
|---|---|---|---|---|---|---|---|
| DNABERT-2 | 8W/8A | 67.33 | 20.74 | 41.80 | 32.98 | 29.08 | 19.58 |
| | 6W/6A | 66.05 | 14.45 | 38.10 | 32.74 | 27.87 | 19.58 |
| | 4W/4A | 2.54 | -0.73 | 0.93 | 2.14 | 6.01 | 0.98 |
| GERM-T | 8W/8A | 72.14 | 46.57 | 50.37 | **35.38** | 24.90 | 23.52 |
| | 6W/6A | 71.47 | 44.40 | 49.26 | **35.16** | 23.22 | 22.87 |
| | 4W/4A | 0.00 | 0.00 | 0.00 | -1.65 | 1.83 | -1.30 |
| GERM | 8W/8A | **72.20** | **50.02** | **53.20** | 33.43 | **32.80** | **24.68** |
| | 6W/6A | **71.52** | **49.17** | **53.10** | 32.14 | **32.92** | **25.61** |
| | 4W/4A | **70.33** | **44.65** | **49.23** | **28.33** | **23.55** | **22.43** |

| Model | Bits | Epigenetic Marks Prediction | | | | Promoter Detection | | |
| | | H3K79me3 | H3K9ac | H4 | H4ac | all | notata | tata |
|---|---|---|---|---|---|---|---|---|
| DNABERT-2 | 8W/8A | 60.11 | 38.01 | 57.40 | 18.80 | 75.24 | 87.94 | 37.51 |
| | 6W/6A | 56.96 | 38.26 | 59.30 | 16.63 | 72.26 | 89.34 | 38.56 |
| | 4W/4A | -0.48 | 2.11 | 0.00 | 0.56 | 0.00 | 5.89 | 0.00 |
| GERM-T | 8W/8A | 61.12 | 49.38 | 78.25 | 44.21 | **82.19** | **91.75** | **59.80** |
| | 6W/6A | **62.14** | 48.76 | 78.25 | 41.23 | 72.25 | **91.75** | 56.87 |
| | 4W/4A | 0.00 | 2.08 | 3.31 | 0.00 | 8.71 | 1.09 | 0.31 |
| GERM | 8W/8A | **61.64** | **49.87** | **79.19** | **48.48** | 79.24 | **92.48** | 57.42 |
| | 6W/6A | 62.01 | **50.08** | **79.33** | **48.36** | **79.33** | **92.13** | 57.42 |
| | 4W/4A | **59.70** | **46.49** | **71.66** | **42.71** | **79.79** | **89.34** | **49.81** |

| Model | Bits | Transcription Factor Prediction (Human) | | | | | Core Promoter Detection | | |
| | | 0 | 1 | 2 | 3 | 4 | all | notata | tata |
|---|---|---|---|---|---|---|---|---|---|
| DNABERT-2 | 8W/8A | 52.40 | 52.31 | 48.78 | 17.02 | **55.88** | **63.98** | 61.25 | 69.12 |
| | 6W/6A | **52.53** | 49.67 | 45.04 | 9.14 | **52.14** | 62.94 | 55.92 | 69.12 |
| | 4W/4A | 5.68 | 11.69 | 11.10 | 2.62 | 0.00 | 4.91 | 7.66 | 16.69 |
| GERM-T | 8W/8A | **60.47** | **82.30** | **73.72** | **68.07** | 44.32 | 63.48 | **66.55** | **72.93** |
| | 6W/6A | 44.96 | 51.87 | **72.70** | 63.35 | 41.84 | **63.72** | **66.55** | **72.93** |
| | 4W/4A | 9.50 | 13.09 | 0.00 | 6.22 | 0.00 | 12.19 | 35.90 | 14.08 |
| GERM | 8W/8A | 45.92 | 54.67 | 56.42 | 67.66 | 39.22 | 48.40 | 64.28 | 68.68 |
| | 6W/6A | 45.18 | **53.28** | 55.45 | **65.81** | 38.44 | 48.37 | 63.60 | 68.68 |
| | 4W/4A | **54.40** | **49.31** | **43.44** | **44.76** | **59.61** | **38.51** | **60.73** | **56.20** |

| Model | Bits | Transcription Factor Prediction (Mouse) | | | | | Virus | Splice |
| | | 0 | 1 | 2 | 3 | 4 | Covid | Reconstruct |
|---|---|---|---|---|---|---|---|---|
| DNABERT-2 | 8W/8A | 48.88 | 64.61 | 62.21 | 65.79 | 41.44 | 46.29 | 63.82 |
| | 6W/6A | **51.31** | **67.05** | 64.04 | 59.92 | 37.17 | 44.42 | 62.13 |
| | 4W/4A | -1.40 | 2.44 | 6.08 | 0.82 | -0.74 | 44.60 | 0.00 |
| GERM-T | 8W/8A | **60.47** | **82.30** | **73.72** | **68.07** | **44.32** | 39.59 | **75.30** |
| | 6W/6A | 44.96 | 51.87 | **72.70** | **63.35** | **41.84** | 39.59 | 68.19 |
| | 4W/4A | 0.00 | 0.00 | 9.05 | 0.00 | 3.53 | -0.18 | 0.00 |
| GERM | 8W/8A | 45.92 | 54.67 | 56.42 | 67.66 | 39.22 | **46.90** | 70.33 |
| | 6W/6A | 45.18 | 53.28 | 55.45 | 65.81 | 38.44 | **47.12** | **70.33** |
| | 4W/4A | **42.15** | **52.01** | 32.18 | **62.06** | **32.30** | **44.60** | **33.59** |

Table 15: **Performance Comparison of Traditional W8A8 PTQ with DNABERT-2.** This table shows the performance of all models on the Traditional W8A8 post-training quantization (PTQ).

| Model | Epigenetic Marks Prediction | | | | | |
|---|---|---|---|---|---|---|
| | H3 | H3K14ac | H3K36me3 | H3K4me1 | H3K4me2 | H3K4me3 |
| DNABERT-2 | 50.39 | 24.73 | 26.09 | 21.80 | 26.80 | 5.05 |
| GERM-T | 63.62 | 29.51 | 39.58 | 26.13 | 19.86 | 17.98 |
| GERM | **70.63** | **50.93** | **53.15** | **33.07** | **35.75** | **24.79** |

| Model | Epigenetic Marks Prediction | | | | Promoter Detection | | |
|---|---|---|---|---|---|---|---|
| | H3K79me3 | H3K9ac | H4 | H4ac | all | notata | tata |
| DNABERT-2 | 48.50 | 42.11 | 70.95 | 3.67 | 71.94 | 60.04 | 34.55 |
| GERM-T | 56.87 | 35.84 | 75.44 | 24.97 | 67.34 | 80.33 | 25.41 |
| GERM | **61.46** | **50.33** | **78.53** | **47.56** | **80.97** | **92.28** | **60.05** |

| Model | Transcription Factor Prediction (Human) | | | | | Core Promoter Detection | | |
|---|---|---|---|---|---|---|---|---|
| | 0 | 1 | 2 | 3 | 4 | all | notata | tata |
| DNABERT-2 | 6.98 | 26.21 | 57.43 | 22.41 | 42.90 | 49.62 | 38.69 | 34.68 |
| GERM-T | 50.73 | 16.09 | 44.11 | 21.43 | 46.94 | **53.68** | 61.83 | 68.45 |
| GERM | **57.11** | **53.20** | **51.67** | **45.65** | **67.63** | 50.85 | 64.42 | 69.35 |

| Model | Transcription Factor Prediction (Mouse) | | | | | Virus | Splice |
|---|---|---|---|---|---|---|---|
| | 0 | 1 | 2 | 3 | 4 | Covid | Reconstruct |
| DNABERT-2 | 21.06 | **65.35** | **65.32** | 29.29 | 11.28 | 2.33 | 14.58 |
| GERM-T | 19.18 | 57.24 | 16.69 | 14.64 | 27.57 | 6.25 | 6.88 |
| GERM | **45.50** | 59.92 | 53.15 | **62.44** | **38.58** | **66.99** | **75.55** |

GERM demonstrates exceptional adaptability and performance in post-training quantization tasks, outperforming both DNABERT-2 and GERM-T across various quantization methods. GERM-T also shows commendable performance, especially in 8-bit quantization, making it a viable alternative when GERM may not be applicable. These models collectively represent significant advancements in deploying efficient and accurate genomic prediction tools in environments with limited computational resources.

## D.4 All Results of Performance Comparison in Resource-Constrained Computing Environments

In this section, we present the results of Performance Comparison in Resource-Constrained Computing Environments. All models were trained on the same computing infrastructure (Nvidia GeForce RTX 2080 Ti 11GB) to ensure a fair comparison. The training time represents the average time per epoch, with OmniQuant used as quantization example.

GERM demonstrates superior adaptability and performance in resource-constrained computing environments compared to DNABERT-2 and GERM-T. Its consistent high MCC scores and reduced training and inference times across various quantization levels and fine-tuning methods establish GERM as the most robust and efficient model, with GERM-T following as a commendable second-best option. These attributes make GERM a promising candidate for further research and application in settings demanding both high performance and computational efficiency.

Table 16: **Comparison of Performance in Resource-Constrained Computing Environments.** Comparison of three models on the quantization and fine-tuning task.

| Method | #Bits | MCC ($\uparrow$) | Time (sec.) |
|---|---|---|---|
| DNABERT-2 | 16W/16A | 59.11 | 7.66 |
| GERM | 16W/16A | **59.73** | **6.70** |
| GERM-T | 16W/16A | 59.30 | 7.01 |
| DNABERT-2 | 8W/8A | 49.92 | 5.47 |
| GERM | 8W/8A | **55.99** | **4.79** |
| GERM-T | 8W/8A | 56.80 | 5.01 |
| DNABERT-2 | 4W/4A | -1.03 | 3.81 |
| GERM | 4W/4A | **20.05** | **3.33** |
| GERM-T | 4W/4A | 0.22 | 3.49 |

| Method | Fine-Tuning Method | MCC ($\uparrow$) | Time (sec.) | |
|---|---|---|---|---|
| | | | Train | Inference |
| DNABERT-2 | Full | 59.11 | 516.49 | 3.85 |
| GERM | Full | **59.73** | **323.10** | **3.24** |
| GERM-T | Full | 59.30 | 326.91 | 3.25 |
| DNABERT-2 | LoRA | 50.91 | 197.13 | 4.12 |
| GERM | LoRA | **57.27** | **154.67** | **3.30** |
| GERM-T | LoRA | 55.60 | 167.76 | 3.32 |
| DNABERT-2 | QLoRA | 50.65 | 206.15 | 5.28 |
| GERM | QLoRA | **53.16** | **164.10** | **4.13** |
| GERM-T | QLoRA | 51.50 | 177.95 | 4.17 |
| DNABERT-2 | LoftQ | 50.76 | 251.37 | 5.77 |
| GERM | LoftQ | **53.11** | **199.58** | **4.52** |
| GERM-T | LoftQ | 51.20 | 220.37 | 4.52 |

**Case Study 2: Performance in CPU-only Computing Environments.** To demonstrate GERM's capability in CPU-only computing environments, we perform performance tests on an 64-core Intel(R) Xeon(R) Gold 6338 CPU @ 2.00GHz with 50GB RAM. We compare GERM's per-epoch training and inference times for the LoRA and QLoRA fine-tuning methods. The results, presented in Table 17, indicate that both GERM and GERM-T achieve shorter fine-tuning times per epoch compared to DNABERT-2, with the only exception being QLoRA when deployed, where the time is slightly longer. QLoRA can be slower than LoRA during inference and fine-tuning due to hardware limitations when bf16 (bfloat16) support is unavailable. QLoRA relies on ultra-low-precision quantization (e.g., 4-bit weights) to reduce memory usage and increase efficiency, which works best on systems that support bf16 or similar mixed-precision operations. However, without bf16 support, these low-precision operations must be emulated by converting back to higher precision, introducing computational overhead. This diminishes the intended speed advantage of QLoRA, potentially making it slower than LoRA on incompatible hardware.

Table 17: **Comparison of Performance in CPU-only Computing Environments.** Comparison of three models on the fine-tuning task.

| Method | Fine-Tuning Method | MCC (↑) | Time (sec.) | |
|---|---|---|---|---|
| | | | Train | Inference |
| DNABERT-2 | LoRA | 50.91 | 808.23 | 29.66 |
| GERM | LoRA | **57.27** | **618.68** | **23.10** |
| GERM-T | LoRA | 55.60 | 674.40 | 23.57 |
| DNABERT-2 | QLoRA | 50.65 | 516.04 | 63.17 |
| GERM | QLoRA | **53.16** | **358.34** | **45.28** |
| GERM-T | QLoRA | 51.50 | 418.13 | 46.91 |

## D.5 Evaluation of Common Genomic Foundation Models

In this section, we conduct the experiments to show the performance of common GFMs. As there is no official fine-tuning code for the Nucleotide Transformer (Dalla-Torre et al., 2024), we utilize its open-sourced checkpoints available on HuggingFace Model Hub [2] and fine-tune it using our code base with LoRA. We implement HyenaDNA (Nguyen et al., 2024b) using its official implementation available on HuggingFace [3] and the HuggingFace Trainer. Since HyenaDNA does not natively support LoRA for fine-tuning, we do not evaluate LoRA's performance on HyenaDNA. As shown in Table 18,

Table 18: **Performance Comparison of Common Genomic Foundation Models.**

| Model | Fine-Tuning Method | MCC | Average Performance Drop |
|---|---|---|---|
| DNABERT-2 | Full | 59.11 | - |
| | LoRA | 50.91 | 13.87% |
| HyenaDNA | Full | 51.31 | - |
| | LoRA | - | - |
| NT-500M-human | Full | 56.05 | - |
| | LoRA | 52.66 | 6.44% |

---

[2]https://huggingface.co/InstaDeepAI

[3]https://huggingface.co/LongSafari/hyenadna-medium-450k-seqlen-hf

## D.6 Performance of GERM on Alternative Transformer-based Models

In this section, we conduct our experiment to validate the effectiveness of the outlier removal approach using alternative transformer-based models, evaluating performance through Matthews Correlation Coefficient (MCC) and average performance drop. We use the NT-500M-human[4] as the target model for our evaluation. Table 19 compares these metrics across NT-500M-human, GERM, and GERM-T models using different low-rank adaptation methods. Table 20 examines the impact of various quantization techniques on the same models. The results demonstrate the effectiveness of outlier removal across diverse adaptation and quantization strategies, highlighting the balance between performance and resource efficiency.

Table 19: **Low-Rank Adaptation Methods Comparison.** This comparison evaluates the performance of different low-rank adaptation methods, including Full, LoRA, QLoRA, and LoftQ, on Nucleotide Transformer 500M models. The best results are highlighted in bold, while the second-best results are underlined.

| Model | Fine-Tuning Method | MCC | Delta MCC | Average Performance Drop |
|---|---|---|---|---|
| NT-500M-human | Full | 56.05 | - | - |
| | LoRA | 52.66 | 3.39 | 6.44% |
| | QLoRA | 51.46 | 4.59 | 8.19% |
| | LoftQ | 51.89 | 4.16 | 7.42% |
| GERM (NT-500M-human) | Full | 55.52 | 0.53 | - |
| | LoRA | 54.32 | 1.73 | **2.16%** |
| | QLoRA | 53.78 | 2.27 | **3.13%** |
| | LoftQ | 54.24 | 1.81 | **2.30%** |
| GERM-T (NT-500M-human) | Full | 56.53 | -0.48 | - |
| | LoRA | 54.89 | 1.16 | 2.90% |
| | QLoRA | 52.78 | 3.27 | 6.63% |
| | LoftQ | 53.45 | 2.60 | 5.45% |

---

[4]https://huggingface.co/InstaDeepAI/nucleotide-transformer-500m-human-ref

Table 20: **Quantization Methods Comparison.** This comparison analyzes the performance of various quantization methods, including FP16, W8A8, Outlier, SmoothQuant, and OmniQuant, on Nucleotide Transformer 500M models. The best results are highlighted in bold, while the second-best results are underlined.

| Model | #Bits | Quantization Method | MCC | Delta MCC | Average Performance Drop |
|---|---|---|---|---|---|
| NT-500M-human | 16W/16A | - | 56.05 | - | - |
| | 8W/8A | - | 34.66 | 21.39 | 38.17% |
| | 8W/8A | Outlier | 32.95 | 23.10 | 41.21% |
| | 6W/6A | | 26.65 | 29.40 | 52.45% |
| | 8W/8A | SmoothQuant | 38.23 | 17.82 | 31.79% |
| | 6W/6A | | 28.67 | 27.38 | 48.84% |
| | 4W/4A | | 3.54 | 52.51 | 93.68% |
| | 8W/8A | OmniQuant | 47.35 | 8.70 | 15.52% |
| | 6W/6A | | 43.63 | 12.42 | 22.16% |
| | 4W/4A | | 5.34 | 50.71 | 90.47% |
| GERM (NT-500M-human) | 16W/16A | - | 55.53 | 0.52 | - |
| | 8W/8A | - | 53.67 | 2.38 | **3.35%** |
| | 8W/8A | Outlier | 45.71 | 10.34 | **17.68%** |
| | 8W/8A | | 41.38 | 14.67 | 25.48% |
| | 8W/8A | SmoothQuant | 53.18 | 2.87 | 4.23% |
| | 6W/6A | | 52.43 | 3.62 | **5.58%** |
| | 4W/4A | | 24.96 | 31.09 | **55.05%** |
| | 8W/8A | OmniQuant | 52.45 | 3.60 | **5.55%** |
| | 6W/6A | | 51.56 | 4.49 | **7.15%** |
| | 4W/4A | | 46.45 | 9.60 | **16.35%** |
| GERM-T (NT-500M-human) | 16W/16A | - | 56.53 | -0.48 | - |
| | 8W/8A | - | 40.71 | 15.34 | 27.99% |
| | 8W/8A | Outlier | 45.98 | 10.07 | 18.66% |
| | 6W/6A | | 43.38 | 12.67 | **23.26%** |
| | 8W/8A | SmoothQuant | 54.19 | 1.86 | **4.14%** |
| | 6W/6A | | 38.67 | 17.38 | 31.59% |
| | 4W/4A | | 10.57 | 45.48 | 81.29% |
| | 8W/8A | OmniQuant | 52.46 | 3.59 | 7.20% |
| | 6W/6A | | 51.34 | 4.71 | 9.18% |
| | 4W/4A | | 23.57 | 32.48 | 58.31% |

## D.7 Performance of GERM on Large-scale GFMs

In this section, we present experiments to validate the effectiveness of GERM on large-scale GFMs. We use the NT-2.5B-multi[5] as the target model for our evaluation. Table 21 compares these metrics across NT-2.5B-multi, GERM, and GERM-T models using different low-rank adaptation methods. Table 22 extends this analysis to evaluate the impact of various quantization techniques on the same models. In the larger-parameter model, we adopt stricter quantization bits. This choice aims to save computation and improve efficiency, as finer compression is crucial when model parameters scale up. Additionally, experiments conducted with a larger-parameter model further validate these findings, demonstrating that outlier removal consistently enhances performance and resource efficiency across diverse adaptation and quantization strategies.

Table 21: **Comparison of Low-Rank Adaptation Methods in Large-Scale Models.** This comparison evaluates the performance of different low-rank adaptation methods, including Full, LoRA, QLoRA, and LoftQ, on Nucleotide Transformer 2.5B models. The best results are highlighted in bold, while the second-best results are underlined.

| Model | Fine-Tuning Method | MCC | Delta MCC | Average Performance Drop |
|---|---|---|---|---|
| NT-2.5B-multi | Full | 56.98 | - | - |
| | LoRA | 53.50 | 3.48 | 6.11% |
| | QLoRA | 52.29 | 4.69 | 8.19% |
| | LoftQ | 52.89 | 4.09 | 7.17% |
| GERM (NT-2.5B-multi) | Full | 57.16 | -0.18 | - |
| | LoRA | 55.98 | 1.18 | **2.06%** |
| | QLoRA | 55.52 | 1.64 | **2.87%** |
| | LoftQ | 55.80 | 1.36 | **2.38%** |
| GERM-T (NT-2.5B-multi) | Full | 56.82 | 0.16 | - |
| | LoRA | 55.24 | 1.58 | 2.78% |
| | QLoRA | 53.32 | 3.50 | 6.16% |
| | LoftQ | 53.74 | 3.08 | 5.42% |

---

[5]https://huggingface.co/InstaDeepAI/nucleotide-transformer-2.5b-multi-species

Table 22: **Comparison of Quantization Methods in Large-Scale Models.** This comparison analyzes the performance of various quantization methods, including FP16, W6A6, W4A4, Outlier, SmoothQuant, and OmniQuant, on Nucleotide Transformer 2.5B models. The best results are highlighted in bold, while the second-best results are underlined.

| Model | #Bits | Quantization Method | MCC | Delta MCC | Average Performance Drop |
|---|---|---|---|---|---|
| NT-2.5B-multi | 16W/16A | - | 56.98 | - | - |
| | 6W/6A | - | 18.52 | 38.46 | 67.50% |
| | 4W/4A | - | 1.39 | 55.59 | 97.56% |
| | 6W/6A | Outlier | 50.23 | 6.75 | 11.85% |
| | 4W/4A | | 40.74 | 16.24 | 28.50% |
| | 6W/6A | SmoothQuant | 47.23 | 9.75 | 17.11% |
| | 4W/4A | | 35.16 | 21.82 | 38.29% |
| | 6W/6A | OmniQuant | 49.55 | 7.43 | 13.04% |
| | 4W/4A | | 43.63 | 13.35 | 23.43% |
| GERM (NT-2.5B-multi) | 16W/16A | - | 57.16 | -0.18 | - |
| | 6W/6A | - | 45.96 | 11.2 | **19.59%** |
| | 4W/4A | - | 42.48 | 14.68 | **25.68%** |
| | 6W/6A | Outlier | 52.24 | 4.92 | 8.61% |
| | 4W/4A | | 49.00 | 8.16 | **14.28%** |
| | 6W/6A | SmoothQuant | 51.95 | 5.21 | **9.11%** |
| | 4W/4A | | 48.15 | 31.09 | 15.76% |
| | 6W/6A | OmniQuant | 52.55 | 4.61 | 8.07% |
| | 4W/4A | | 49.26 | 7.90 | **13.82%** |
| GERM-T (NT-2.5B-multi) | 16W/16A | - | 56.82 | 0.16 | - |
| | 6W/6A | - | 32.58 | 24.24 | 42.66% |
| | 4W/4A | - | 10.49 | 46.33 | 81.54% |
| | 6W/6A | Outlier | 52.14 | 4.68 | **8.24%** |
| | 4W/4A | | 46.24 | 10.58 | 18.62% |
| | 6W/6A | SmoothQuant | 51.61 | 5.21 | 9.17% |
| | 4W/4A | | 48.12 | 8.70 | **15.31%** |
| | 6W/6A | OmniQuant | 52.43 | 4.39 | **7.73%** |
| | 4W/4A | | 47.28 | 9.54 | 16.79% |

## D.8 Performance of GERM with Different Outlier Removal Techniques

In this section, we present experiments to validate the effectiveness of GERM compared to other common outlier removal techniques, including clipped softmax and gated attention (Bondarenko et al., 2024). We utilize SmoothQuant as the quantization method in our experiments. As shown in Table 23, the results demonstrate the superior effectiveness of outlier removal by GERM compared to clipped softmax and gated attention. GERM achieves a performance improvement of 2.59% in 4-bit quantization and 1.99% in 8-bit quantization.

Table 23: **Performance of GERM with Clipped Softmax and Gated Attention**

| Method | #Bits | MCC ($\uparrow$) | Average Performance Drop |
|---|---|---|---|
| DNABERT-2 | 16W/16A | 59.11 | - |
| GERM | 16W/16A | 59.73 | - |
| Clipped Softmax | 16W/16A | 59.17 | - |
| Gated Attention | 16W/16A | 59.49 | - |
| DNABERT-2 | 8W/8A | 49.92 | 15.55% |
| GERM | 8W/8A | **55.99** | **6.26%** |
| Clipped Softmax | 8W/8A | 53.26 | 9.99% |
| Gated Attention | 8W/8A | 54.58 | 8.25% |
| DNABERT-2 | 4W/4A | -1.03 | 101.74% |
| GERM | 4W/4A | **20.05** | **66.43%** |
| Clipped Softmax | 4W/4A | 13.49 | 77.20% |
| Gated Attention | 4W/4A | 18.66 | 69.02% |

## D.9 PTQ Performance of GERM Following LoRA Fine-Tuning

In this section, we present experiments to validate the effectiveness of GERM on post-training quantization (PTQ) performance following LoRA adaptation. We employ SmoothQuant as the quantization method in our evaluation. As shown in Table 24, the results highlight the superior outlier mitigation capability of GERM, leading to improved PTQ performance. Specifically, using W8A8 quantization GERM achieves a 84.72% improvement over the baseline, demonstrating its effectiveness in low-rank adaptation and quantization scenarios.

Table 24: **Comparison of Post-Training Quantization (PTQ) Performance Across Low-Rank Adaptation Methods in Large-Scale Models** This comparison assesses the performance of SmoothQuant (SQ) following low-rank adaptation fine-tuning using LoRA on DNABERT-2 117M models. The top-performing results are shown in bold, and the second-best results are underlined for clarity.

| Model | Method | MCC | Delta MCC | Average Performance Drop |
|---|---|---|---|---|
| | Full | 59.11 | 7.00 | - |
| DNABERT-2 | LoRA | 50.91$\pm$1.67 | 15.2 | 13.87% |
| | LoRA + SQ | 34.23$\pm$1.56 | 31.88 | 42.09% |
| | Full | 59.73 | 6.38 | - |
| GERM | LoRA | 57.27$\pm$0.70 | 8.84 | **4.12%** |
| | LoRA + SQ | 55.89$\pm$0.93 | 10.22 | **6.43%** |
| | Full | 59.30 | 6.81 | - |
| GERM-T | LoRA | 55.60$\pm$0.28 | 10.51 | 6.23% |
| | LoRA + SQ | 54.26$\pm$0.65 | 11.85 | 8.50% |

### D.10 Attention Distribution Visualization in GERM

In this section, to better understand the impact of outlier removal in GERM, we visualize the attention score distributions across transformer layers, as shown in Figure 3. We visualize the attention distributions for three different DNA sequences from the mouse 0 dataset, showing both attention probabilities and attention scores for the vanilla and GERM versions of DNABERT-2. The attention probability captures the probability between each token and all other tokens, while the attention score is computed by multiplying the attention probability with the attention value, yielding a tensor of shape (number of tokens × hidden dimension). For visualization purposes, we use the last hidden layer of the model and display the first 32 dimensions of the attention score. These visualizations reveal that GERM produces smoother and more stable attention patterns compared to baseline models such as DNABERT-2, which exhibit sharp spikes and irregularities due to the influence of outliers. The reduced variance and kurtosis in GERM's attention maps confirm its ability to suppress low-information tokens, resulting in more efficient and interpretable attention behavior.

## E Additional Discussion and Limitation

In this section, we provide additional discussion on the limitations of GERM-T. As a trade-off approach, GERM-T is designed as a practical extension of GERM for continual low-resource adaptation, minimizing recomputation by reusing pre-trained checkpoints and applying small-step updates. While it achieves a balance between performance and computational cost under 8-bit quantization, one key limitation is that GERM-T performs worse in certain quantization and low-rank adaptation settings, particularly under low-bit quantization scenarios. This performance degradation results primarily from the restricted optimization scope imposed by small-step fine-tuning and the accumulation of approximation errors. Comparing with GERM, these errors prevent GERM-T from fully mitigating inherent outliers, leading to increased *average kurtosis* and *maximum infinity norm*. In future work, we explore the underlying causes of this degradation in greater depth and develop a more robust quantization-aware training (QAT) approach to better manage these trade-offs.

## F Definition of Outlier in Our Paper

In this section, we provide more detailed deification of outlier in our paper. In our work, we define **outliers** as *tokens or activations that disproportionately influence the attention mechanism*, despite containing little or no meaningful information. These outliers emerge when the softmax function amplifies the attention probabilities of tokens that ideally should receive minimal or zero focus. We use the following attention mechanism to analyze this behavior:

$$\text{Output} = \text{Residual}\left(\text{Softmax}\left(\frac{QK^\top}{\sqrt{d}}\right)V + X\right).$$

As shown in Hu et al. (2024a), if the attention input $X$ already contains sufficient information, the attention mechanism within the residual connection should ideally behave like an *identity transform*, producing near-zero attention outputs:

$$\text{Softmax}\left(\frac{QK^\top}{\sqrt{d}}\right)V \approx 0.$$

In such cases, tokens with high values in $V$—which may represent biologically significant features—should still receive near-zero attention probabilities.

**Why Classic Softmax Fails.** The problem arises from how the softmax function normalizes probabilities. Softmax enforces that all probabilities sum to one, which inherently magnifies the attention probabilities assigned to *low-value tokens*. This unwanted amplification broadens the attention score distribution and introduces **outliers**—tokens that exert disproportionate influence despite their low informational value.

These outliers are particularly problematic in genomic models like DNABERT-2, where certain regions of genomic sequences—such as repetitive patterns, low-complexity sequences, or non-coding regions—resemble no-op tokens. While these regions carry minimal biological relevance, the classic softmax mechanism may assign them higher-than-expected attention scores, diverting focus away from meaningful genomic features.

**Intuition Behind Outliers in Genomic Models.**    Outliers in genomic models typically arise from sequence patterns that produce anomalous query-key interactions in the attention mechanism. While genomic data lacks traditional "words" as in language models, certain biological patterns exhibit similar behavior. Key examples include:

- **Low-Complexity Regions (e.g., Poly-A or Poly-T Sequences):** Genomic sequences often include regions with runs of identical bases (e.g., `AAAAA...`, `TTTTT...`). These sequences contain minimal unique information yet can produce large, uniform dot-product values in the attention mechanism. This causes softmax to assign exaggerated probabilities to these low-information tokens, effectively making them outliers.

- **Repetitive Motifs and Tandem Repeats:** Certain genomic regions, such as microsatellites and tandem repeats, involve recurring nucleotide patterns that behave similarly to low-value tokens in the attention mechanism. These patterns exhibit strong internal correlations, often resulting in softmax overemphasizing them as if they were biologically significant.

- **Boundary and Spacer Elements (e.g., Alignment Padding or Non-coding Spacer Sequences):** In genomic datasets, artificial padding sequences, non-coding segments, or spacer sequences are sometimes introduced to ensure proper sequence alignment. These tokens are intended to have no biological relevance, yet softmax's behavior inadvertently amplifies their attention scores, creating noise that distorts meaningful patterns.

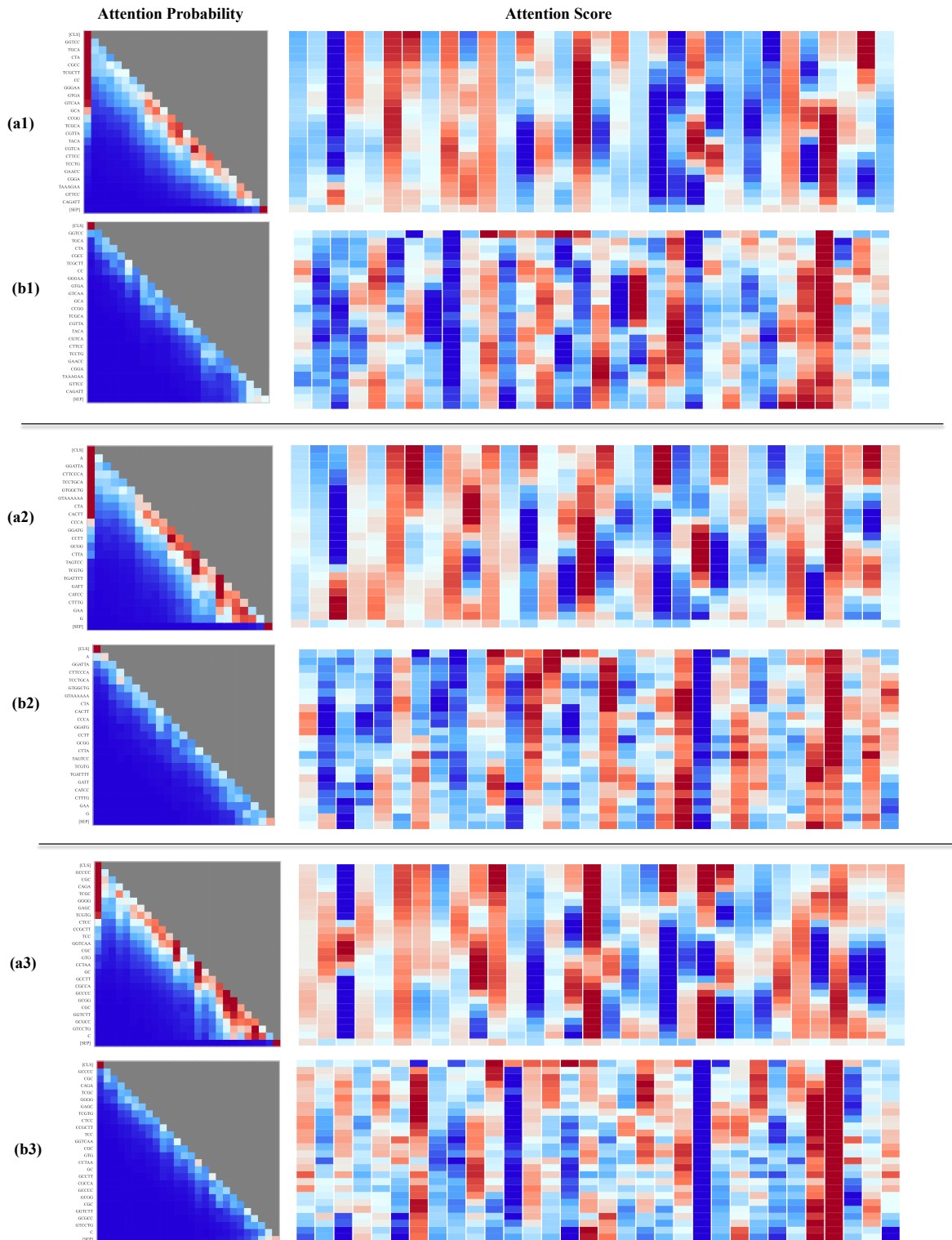

Figure 3: **Attention Distribution Visualization in GERM.** Comparison of attention probabilities and outputs for a genomic sample between DNABERT-2 and NT-500M-human. Heatmaps from the final hidden layers are scaled from 0 (blue) to 1 (red). In the figure, all rows labeled (a) correspond to the vanilla DNABERT-2, while all rows labeled (b) represent the GERM version. The vanilla model exhibits a broad attention spread, which dilutes focus across tokens, whereas GERM concentrates attention on key tokens, enhancing both efficiency and interpretability.

