# OpenReview forum: "Fast and Low-Cost Genomic Foundation Models via Outlier Removal"
_ICML.cc/2025/Conference — ICML 2025 poster_

### Official Review · Reviewer_8UsM · 2025-03-10

**Overall Recommendation:** 4

**Summary:**

The paper "Making Genomic Foundation Models more Foundational Requires Outlier Removal: A Case Study on DNABERT-2" introduces GERM, an outlier-free genomic foundation model (GFM) designed to improve quantization robustness and low-rank adaptation efficiency. The authors argue that eliminating outliers in attention mechanisms significantly enhances computational efficiency and model performance under resource constraints. Empirical results demonstrate substantial improvements in quantization and fine-tuning metrics compared to DNABERT-2. While the work addresses critical challenges in deploying GFMs, several methodological and experimental aspects require clarification and validation.

**Claims And Evidence:**

1. The claim that GERM reduces computational costs and improves quantization robustness is supported by experiments showing a 92.14% reduction in average kurtosis and 82.77% in maximum infinity norm. Also, the authors validate the connection between outlier metrics and practical deployment, such as inference time in resource-constrained environments.
2. The authors demonstrate that GERM outperforms DNABERT-2 by 37.98% in fine-tuning and 64.34% in quantization, validating the effectiveness of the proposed "outlier-free" method.
3. The small-step continual learning strategy (GERM-T) avoids retraining from scratch, which is pragmatic. However, its performance degradation in 4-bit quantization (Table 1) suggests limitations in outlier mitigation for extreme compression.

**Essential References Not Discussed:**

There are some other GFMs should include in literature review like Evo which Integrates Hyena operators for efficient sequence modeling.

**Experimental Designs Or Analyses:**

The experimental design does not entirely make sense to me. I have several concerns:

**LoRA Experiments:** I would also suggest to separate the experiments to two fold, if LoRA+Quantization is one of the focus on this work. In a first experiment, we can investigate the impact of LoRA (with FP 32/16)

**Model Size**: The largest GFM used in this paper is NT-2.5B. If there are larger models available, I recommend that the authors include them in the experiments to provide a more comprehensive evaluation.

**Methods And Evaluation Criteria:**

**Outlier-Free Attention**: Replacing Softmax with a modified Hopfield layer is innovative, but it lacks a detailed comparison to other outlier suppression techniques mentioned in supplementary material, such as clipped attention and gated mechanisms.

**Other Comments Or Suggestions:**

I do not have additional comments or suggestions

**Other Strengths And Weaknesses:**

**Strengths**:

- Novel integration of outlier-free attention with ALiBi for variable-length sequences.
- Comprehensive evaluation across multiple quantization methods and adaptation strategies.

**Weaknesses**:

- Unclear practical impact of outlier metrics, questions like how kurtosis reduction translates to real-world deployment gains need further discussion.
- Incomplete discussion of limitations, particularly GERM-T’s performance trade-offs.

**Questions For Authors:**

Questions are already listed above. I have no further questions

**Relation To Broader Scientific Literature:**

This work aligns with efforts to optimize transformer-based models for resource-constrained settings, such as QLoRA and SmoothQuant. The authors also discuss state-of-the-art models like HyenaDNA and NT-2.5B.

**Theoretical Claims:**

1. Definition of the outlier in the context of the paper. Especially considering that classical definition of outlier in statistics and ML is that observation is not from the distribution that is being modeled.

2. The theoretical analysis in Appendix B assumes non-singular weight matrices and ideal low-rank adaptation conditions. While mathematically sound, these assumptions may not hold for real-world GFMs with complex parameter interactions.

---

> ### Author Rebuttal · Authors · 2025-03-31
>
> The updated manuscript can be accessed anonymously at [link](https://www.dropbox.com/scl/fi/itpm5n21pfu3at01bofab/germ_icml2025.pdf?rlkey=zl8noukikpz1s4493b752s9uj&e=1&st=qma9ihc9&dl=0).
> > **Reviewer's Comment**: a detailed comparison to ...
>
> **Response**: We thank the reviewer for the feedback and the opportunity to clarify our evaluation of outlier suppression techniques. We would like to highlight that we have already included a detailed comparison of GERM with alternative outlier suppression methods in **Table 23** and **Appendix E.8** of the supplementary material. We are happy to provide further clarification if needed.
> > **Reviewer's Comment**: Definition of the outlier…
>
> **Response**: We appreciate the opportunity to clarify the definition of **outliers** in our work and how it differs from the classical statistical definition. A more detailed explanation is provided in the response to the reviewer `AD8a`.
> > **Reviewer's Comment**: The theoretical analysis in Appendix B assumes…
>
> **Response**: Thank you for the helpful feedback. We acknowledge that our assumptions may not hold universally. However, we believe they remain practical in real-world GFMs for three reasons:
> 1. Weight matrices in large foundation models rarely become singular. Overparameterization and regularization typically prevent this.
>
>
> 2. The low-rank assumption may not hold in every scenario. Yet it helps us show the strong expressiveness of LoRA tuning transformers with outlier-free layers. Many theoretical analyses use similar assumptions, so they form a reasonable setup.
>
>
> 3. Under these conditions, our results show that LoRA-tuned outlier-free transformers match or exceed the expressiveness of softmax-based architectures. These assumptions may not apply everywhere, but they do not undermine our theoretical insights.
>
> > **Reviewer's Comment**: LoRA Experiments...
>
> **Response**: We appreciate the reviewer’s suggestion to disentangle the effects of LoRA and quantization for a more granular analysis. In our revised manuscript, we add additional experiments and analysis for LoRA+Quantization in  **Table 24 and Appendix E.9**.
>
> > **Reviewer's Comment**: larger models available?...
>
> **Response**: Thank you for your advice to include larger models for a more comprehensive evaluation. In our study, we select **NT2.5B** as the largest model for the classification task because it represents the most suitable model architecture designed for genomic sequence classification.
> While larger models such as **Evo** and **GeneOcean** exist, they are fundamentally designed as generation models rather than classifiers. These models prioritize sequence generation capabilities rather than directly optimizing for classification accuracy. As a result, they differ significantly in architecture, objective function, and training strategy. So, making a direct comparison with GERM in a classification context is less appropriate.
> > **Reviewer's Comment**: Unclear practical impact of outlier metrics...
>
> **Response**: Thank you for the comment. We clarify that the outlier metrics, such as kurtosis and max infinite norm, are empirically shown to correlate with model quantizability—that is, robustness to performance degradation under quantization [1,3]. Prior studies [2,4] demonstrate that outliers significantly reduce quantized model performance, and thus reducing these metrics has direct implications for improved performance of quantization.
>
> [1] Bondarenko, et al. "Quantizable transformers"
>
> [2] Wei, Xiuying, et al. "Outlier suppression"
>
> [3] Chmiel, Brian, et al. "Robust quantization"
>
> [4] Dettmers, Tim, et al. "Gpt3. int8 ()"
>
> > **Reviewer's Comment**: other GFMs should include in literature review...
>
> **Response**: We appreciate the reviewer's suggestion to expand our literature review by including additional GFMs. In response, we incorporate Evo in the related work section of our revised version.
>
> > **Reviewer's Comment**: Incomplete discussion of limitations?...
>
> **Response**: We thank the reviewer’s suggestions regarding the need for a more complete discussion of GERM-T’s limitations. We update the discussion of GERM-T's limitations in our revised manuscript, specifically in **Appendix F**.

---

> > ### Comment · Reviewer_8UsM · 2025-04-02
> >
> > Thank you for your response and the insightful experiments. The authors have addressed most of my concerns, and I have accordingly increased my original score.

---

> > > ### Author Response · Authors · 2025-04-02
> > >
> > > We are very pleased to have addressed your concerns and thank you very much for raising the score!

---

### Official Review · Reviewer_PKFp · 2025-03-13

**Overall Recommendation:** 4

**Summary:**

This article introduces the outlier-free Hopfield layer into the Genomic foundation model to achieve a better trade-off between performance and efficiency. They also propose a continued training approach to avoid the additional cost of training from scratch. Comprehensive experimental results demonstrate that the outlier-free design significantly reduces performance degradation during quantization or fine-tuning.

## update after rebuttal
I have read the authors' rebuttal and my concerns have been addressed.

**Claims And Evidence:**

The article does not clearly explain what an outlier is or how to quantify it. I suggest that the authors provide quantitative or qualitative evidence to demonstrate that introducing the outlier-free structure can indeed mitigate the outlier phenomenon.

**Essential References Not Discussed:**

NA

**Experimental Designs Or Analyses:**

Yes, the experimental design here includes a solid and thorough ablation study.

**Methods And Evaluation Criteria:**

Yes, they have conducted comprehensive evaluations on various baselines, including downstream fine-tuning and quantization, to support their conclusions.

**Other Comments Or Suggestions:**

NA

**Other Strengths And Weaknesses:**

NA

**Questions For Authors:**

NA

**Relation To Broader Scientific Literature:**

Their research contributes to improving the accessibility of genomic foundation models by providing the community with a more lightweight yet efficient model, which helps accelerate scientific research within the community.

**Theoretical Claims:**

NA

---

> ### Author Rebuttal · Authors · 2025-03-31
>
> The updated manuscript can be accessed anonymously at [link](https://www.dropbox.com/scl/fi/itpm5n21pfu3at01bofab/germ_icml2025.pdf?rlkey=zl8noukikpz1s4493b752s9uj&e=1&st=qma9ihc9&dl=0).
> > **Reviewer's Comment**: does not clearly explain what an outlier is or how to quantify it...
>
> **Response**: We thank the reviewer for the helpful comments and the opportunity to clarify our definition of outliers and provide theoretical analysis for their mitigation.
>
> ## **Definition of Outliers in Our Work**
>
> The definition of **outliers** in our work differs from the classical statistical definition. In traditional statistics and machine learning, outliers are typically defined as data points that fall outside the modeled distribution. In contrast, in our context, we define **outliers** as **tokens or activations that disproportionately influence the attention mechanism**, despite containing little or no meaningful information. A more detailed explanation is provided in the response to the reviewer `AD8a`.
>
> ## **Q2: How Does softmax_1 Mitigate Outliers?**
>
>
>
> In our paper, we introduce **Softmax1** equation, a modified softmax function designed to mitigate the effects of outliers:
>
>
>
> $$
> \text{Softmax1}(S)_i = \frac{\exp(S_i)}{1 + \sum_j \exp(S_j)}
> $$
>
>
>
> Where $\( S = QK^\top / \sqrt{d} \)$ represents the scaled dot product in the attention mechanism.
>
>
>
> **Key Improvements in Softmax1:**
>
>
>
>
> 1. **Suppression of Low-Value Tokens:**
>
> Unlike standard softmax, which assigns **non-zero probabilities** to all tokens — even those with highly negative scores — Softmax1 allows low-information tokens to receive **near-zero probabilities**. This behavior is crucial in genomic models where repetitive sequences or spacer elements resemble no-op tokens.
>
>
>
> 2. **Controlled Attention Distribution:**
>
> Softmax1 suppresses the broadening of the attention distribution, ensuring that the model remains focused on biologically relevant regions rather than noisy patterns.
>
>
>
> ---
>
>
>
> ## **Theoretical Support for Softmax1’s Outlier Resistance**
>
> - **Standard Softmax Behavior:**
>
> Standard softmax assigns non-zero probabilities to all tokens, even those that ideally should receive no attention. In extreme cases:
>
>
>
> $$
> \lim_{x_1 \to -\infty} \ldots \lim_{x_k \to -\infty} \text{Softmax}(x)_i = \frac{1}{k} > 0
> $$
>
>
>
> - **Softmax1 Behavior:**
>
> In contrast, Softmax1 ensures that tokens with highly negative scores are assigned probabilities that collapse to zero:
>
>
>
> $$
> \lim_{x_1 \to -\infty} \ldots \lim_{x_k \to -\infty} \text{Softmax1}(x)_i = 0
> $$
>
>
>
> This limiting behavior ensures that low-information tokens, such as repetitive motifs or spacer tokens, are effectively ignored, stabilizing the attention mechanism.
>
>
>
> ---
>
>
>
> ## **Empirical Evidence for Outlier Mitigation**
>
> To validate that Softmax1 effectively mitigates outliers, we provide both **quantitative** and **qualitative** evidence:
>
>
>
> - **Quantitative Evidence:**
>
> As shown in Table 1, GERM achieves a **92.14% reduction in kurtosis** and an **82.77% reduction in the maximum infinity norm** compared to DNABERT-2. These reductions demonstrate that GERM effectively suppresses extreme values associated with outliers.
>
>
>
> - **Qualitative Evidence (Attention Distribution Plots):**
>
> Visualizations in **Appendix E.10** in revision illustrate that DNABERT-2 exhibits sharp, irregular attention spikes corresponding to outlier tokens, while GERM maintains a smoother, more stable attention distribution.
>
> ---
>
> ## **Conclusion**
>
> We appreciate the opportunity to clarify our definition of outliers and their impact on genomic foundation models. Our theoretical analysis highlights how the attention mechanism illustrates the formation of outliers, while the Softmax1 equation mitigates their influence by reducing the amplification of low-information tokens. The combination of improved theoretical design and strong empirical evidence reinforces GERM’s ability to suppress outliers, enhancing its stability and performance in genomic modeling tasks.

---

### Official Review · Reviewer_AD8a · 2025-03-19

**Overall Recommendation:** 4

**Summary:**

This paper describes an adaptation of the DNABERT genomic foundation model to reduce the impact of outliers in attention mechanism. The outlier phenomenom was first observed in large language models, where it was shown attention mechanisms can learn to pay large attention to irrelevant tokens, like the [SEP] token. This is believed to stem from cases where an attention head needs something equivalent to a no-op, where it does not attend anywhere. These outliers manifest themselves as certain embeddings in each layer having large magnitude and cause problems for quantization, due to the resulting large range of values to be quantized. Quantization is a key component of making genomic foundation models practical, in the sense that they could be deployed in APIs for non-computational scientists to use in exploratory research.

The authors incorporate the outlier-free Hopfield layer of (Hu et al., 2024a) into the DNABERT model and demonstrate that this reduces the impact of outliers and retains better performance after quantization than the original BERT model. They also demonstrate that this model has better performance after low-rank adaptation and provide an efficient continual learning approach that enables the incorporation of this layer in a pretrained model without retraining from scratcn, and that the performance of this version falls between the GERM model (outlier-free Hopfield layer retrained from scratch) and the original DNABERT model on almost all evaluations.

**Claims And Evidence:**

The authors claims, as described above, are demonstrated with a comprehensive set of experiments.

**Essential References Not Discussed:**

To the best of my knowledge, the authors discuss all the relevant research.

**Experimental Designs Or Analyses:**

The experiments look sound to me.

**Methods And Evaluation Criteria:**

Ther authors use the same set of benchmarks as previous genomic foundation models.

**Other Comments Or Suggestions:**

n/a

**Other Strengths And Weaknesses:**

The paper is well-written, with clear motivation and clearly described experiments. As mentioned above, the contributions are useful and carefeully evaluated.

One possible criticism is that the paper combines ideas that have previously been demonstrated for large language models, rather than providing a technical innovation in itself. However, I think the paper makes a solid contribution as a piece of empirical work others can bulld on.

**Questions For Authors:**

While we have a good explanation of the source of outliers in language models, it wasn't obvious to me what kind of input feature or token would lead to an outlier in a genomic foundation model. Do the authors have an intuition for this?

**Relation To Broader Scientific Literature:**

This work is highly relevant to the broader scientific literature, given the increased interest in genomic foundation models and the likely requirement to use techniques like quantization and low-rank adaptation to make their use practical.

**Theoretical Claims:**

I did not check the theoretical claims (in the appendix) in detail.

---

> ### Author Rebuttal · Authors · 2025-03-31
>
> The updated manuscript can be accessed anonymously at [link](https://www.dropbox.com/scl/fi/itpm5n21pfu3at01bofab/germ_icml2025.pdf?rlkey=zl8noukikpz1s4493b752s9uj&e=1&st=qma9ihc9&dl=0).
> > **Reviewer's Comment**: what would lead to an outlier in a GFM...
>
>
> **Response**: Thank you for your insightful question about the nature of outliers in genomic foundation models. In our work, we define **outliers** as **tokens or activations that disproportionately influence the attention mechanism**, despite containing little or no meaningful information. These outliers emerge when softmax amplifies the attention probabilities of tokens that ideally should receive minimal or zero focus. We use the following attention mechanism to analyze this behavior:
> $$
> \text{Output} = \text{Residual} \left( \text{Softmax} \left( \frac{Q(K)^\top}{\sqrt{d}} \right) V + X \right)
> $$
> As shown in (Hu et al., 2024), if the attention input \(X\) already contains sufficient information, the attention mechanism within the residual connection should ideally behave like an **identity transform**, producing near-zero attention outputs:
> $$
> \text{Softmax} \left( \frac{QK^\top}{\sqrt{d}} \right) V \approx 0
> $$
> In such cases, tokens with **high values in \(V\)** — which may represent biologically significant features — should still receive **near-zero attention probabilities**.
>
>
> ## Why Classic Softmax Fails
>
>
> The problem arises from how the softmax function normalizes probabilities. Softmax enforces that probabilities sum to 1, which inherently magnifies the attention probabilities assigned to **low-value tokens**. This unwanted amplification broadens the attention score distribution and introduces **outliers** — tokens that exert a disproportionate influence despite their low information value.
> Genomic models such as DNABERT-2 face a critical challenge from outliers - including repetitive elements, low-complexity regions, and non-coding segments - similar to no-op tokens. Despite their limited biological significance, these regions receive disproportionately high attention weights through standard softmax operations, consequently suppressing the model's focus on biologically relevant genomic features.
>
>
> ## Intuition Behind Outliers in Genomic Models
>
>
> Outliers in genomic models typically arise from sequence patterns that produce anomalous query-key interactions in the attention mechanism. Though genomic data lacks traditional "words" like language models, certain biological patterns produce similar effects. Key examples include:
> ### 1. **Low-Complexity Regions (e.g., Poly-A or Poly-T Sequences)**
> Genomic sequences frequently contain repetitive base patterns (e.g., `AAAAA...`, `TTTTT...`). These sequences carry minimal unique information yet can produce large, uniform dot-product values in the attention mechanism. This causes softmax to assign exaggerated probabilities to these low-information tokens, effectively making them outliers.
> ### 2. **Repetitive Motifs and Tandem Repeats**
> Genomic repetitive elements such as microsatellites and tandem repeats, which contain patterns (e.g., `(CA)n`, `(GAA)n` repeats) that generate artificially inflated attention scores due to their inherent self-similarity. However, such regions lack corresponding biological information, often resulting in softmax overemphasizing them as if they were biologically significant.
> ### 3. **Boundary and Spacer Elements (e.g., Alignment Padding or Non-coding Spacer Sequences)**
> In genomic datasets, artificial padding sequences, non-coding segments, or spacer sequences are sometimes introduced to ensure proper sequence alignment. These tokens are intended to have no biological relevance, yet softmax’s behavior inadvertently amplifies their attention scores, creating noise that distorts meaningful patterns.
> ## Impact on Genomic Analysis
> In genomic foundation models like DNABERT-2, these outliers negatively impact performance by:
> - **Increasing Error Rates:** Outliers divert attention away from biologically meaningful regions, reducing prediction accuracy for tasks like mutation site identification.
> - **Destabilizing Fine-Tuning:** During fine-tuning, excessive focus on low-information tokens increases noise in gradient updates, limiting convergence stability.
> - **Masking Important Features:** Outliers may overshadow rare but critical genomic patterns, reducing the model’s capacity to detect subtle but meaningful biological signals.
>
> > **Reviewer's Comment**:.One possible criticism is….
>
> **Response**: We appreciate the reviewer’s feedback on the importance of innovation. We also appreciate the reviewer's comments, acknowledging that our paper makes a solid contribution as an empirical study that others can build upon. This paper presents experimental evidence demonstrating that outlier-free methods offer an efficient solution for improving low-rank adaptation and quantization. We restate our contribution in the response to the reviewer `sdac`.

---

### Official Review · Reviewer_sdac · 2025-03-21

**Overall Recommendation:** 2

**Summary:**

This paper addresses the limitations of current GFMs, particularly DNABERT-2, when applying low-bit quantization and parameter-efficient fine-tuning methods like LoRA. The authors attribute performance degradation to outliers in attention mechanisms and propose GERM, a variant using an outlier-free attention mechanism (softmax₁). They also present GERM-T, a continual learning-based model for adapting GFMs under resource constraints. Experiments show that GERM and GERM-T perform better than baselines under low-precision and constrained environments.

**Claims And Evidence:**

Some central claims, such as the harmful effects of outliers in attention distributions on quantization and LoRA performance, are not fully supported with direct quantitative evidence. While statistics like kurtosis and max norm are provided, detailed distributional plots or causal analysis connecting these outliers to downstream performance degradation are lacking. The claim of improved efficiency through outlier removal is empirically supported but not rigorously motivated.

**Essential References Not Discussed:**

The authors appropriately cite related work on outlier-free transformers and GFMs including DNABERT, DNABERT-2, HyenaDNA and Nucleotide Transformer. Further reference that could enhance the current reference would be Evo [1].

[1] Sequence modeling and design from molecular to genome scale with evo

**Experimental Designs Or Analyses:**

Experiments are well-structured and provide comparisons across multiple quantization settings. However, the motivation—attention outliers—is not directly evaluated through visualization or per-layer analysis of attention score distributions. This gap undermines the connection between hypothesis and results.

**Methods And Evaluation Criteria:**

The evaluation framework is reasonable, employing multiple benchmarks (e.g., variant effect prediction, promoter identification) and standard metrics like MCC. The methods, such as GERM and GERM-T, are clearly described and evaluated under realistic constraints (e.g., low-precision hardware). However, the lack of detailed analysis on where and how outliers hurt model behavior weakens the methodological clarity.

**Other Comments Or Suggestions:**

Some useful comments:
1. Clarify how “outliers” are defined and provide attention distribution visualizations.
2. Consider evaluating biological interpretability or downstream relevance more thoroughly.

**Other Strengths And Weaknesses:**

- **Strengths**: Practical focus on resource-limited settings; strong empirical results under quantized and efficient setups.

- **Weaknesses**: Limited novelty (reuses existing attention method), weak quantitative motivation, and missing connection between proposed changes and biological interpretability.

**Questions For Authors:**

1. Can you provide attention score distributions (e.g., histogram or density plots) to support the outlier hypothesis?
2. What is the operational definition of an “outlier” in your analysis (e.g., percentile-based, norm threshold) and how it relates and impacts the biological findings?
3. Given that softmax₁ is not novel, what do you consider the core technical contribution of this work?

**Relation To Broader Scientific Literature:**

The paper builds on DNABERT-2 and outlier mitigation in transformers (e.g., softmax₁ from Hu et al. 2024). While its application to genomics is relevant and timely, the primary technique has already appeared in prior literature, reducing the novelty of the proposed approach.

**Theoretical Claims:**

Supplementary material (Section A) includes theoretical analysis on the softmax₁ function, particularly proving that it produces attention distributions with a bounded second moment, which helps prevent outlier values. While the result is cited from prior work (Hu et al., 2024), the paper correctly restates the theoretical guarantees that motivate the use of softmax₁. No new proofs are introduced, but the prior claims are accurately presented.

---

> ### Author Rebuttal · Authors · 2025-03-31
>
> The updated manuscript can be accessed anonymously at [link](https://www.dropbox.com/scl/fi/itpm5n21pfu3at01bofab/germ_icml2025.pdf?rlkey=zl8noukikpz1s4493b752s9uj&e=1&st=qma9ihc9&dl=0).
>
>
> > **Reviewer's Comment**: However, the motivation ….
>
>
> **Response**: We thank the reviewer for suggesting a direct attention score analysis to validate the outlier hypothesis.
> To address this point, we add detailed visualizations of the attention score distributions in **Figure 3 and Appendix E.10**, where we compare DNABERT-2 and GERM. The results demonstrate that GERM suppresses attention outliers, leading to more focused and efficient attention patterns. This analysis provides direct evidence supporting our hypothesis.
> We clarify this connection in the revised manuscript and invite the reviewer to examine Figure 3 and Appendix E.10 for a detailed analysis of these findings.
>
>
> > **Reviewer's Comment**: While statistics like ..
>
>
> **Response**: We sincerely appreciate the reviewer’s insightful suggestion about the need to more rigorously establish the connection between attention outliers and downstream performance degradation. In response, we have added detailed attention score distribution visualizations in Appendix E.10 and Figure 3, including per-layer heatmaps comparing DNABERT-2 and GERM.
> For a formal definition of “outliers”, we respectfully refer the reviewer to our response to Reviewer `AD8a`. We also discuss in our response to Reviewer `8UsM` how existing literature has demonstrated the negative impact of attention outliers on model performance.
> We believe these additions strengthen the theoretical and empirical motivation of our work and clarify the causal relationship between attention outliers and performance.
>
>
> > **Reviewer's Comment**: Limited novelty…
>
>
> **Response**: We appreciate the reviewer’s feedback on the importance of novelty. While our approach builds upon established methodologies, we would like to highlight our key innovation: we are the first to integrate outlier removal to simultaneously enable **(1) robust quantization and (2) accelerated low-rank adaptation** for genomic foundation models. This contribution is important because genomic data presents unique challenges—such as extreme sparsity, high variability, and frequent outliers in attention mechanisms—which differ significantly from those in traditional NLP.
>
>
> 1. **First to achieve accelerated LoRA for Genomic Models via Systematic Outlier Removal**: Our work pioneers the use of LoRA in genomic foundation models like DNABERT-2. Applying LoRA directly to DNABERT-2 results in performance degradation due to genomic data-specific outliers. This novel integration of the outlier-free Hopfield mechanism enables effective low-rank adaptation alongside robust quantization, achieving a 37.98% improvement in fine-tuning performance compared to DNABERT-2.
> 2. **Adapting Techniques for Genomic Challenges**: The genomic domain demands unique adaptations due to sparse and highly variable tokenization methods like k-mer and BPE. The outlier-free Hopfield layer required significant adjustments to mitigate domain-specific outliers, reducing kurtosis and infinity norm values by 92.14% and 82.77%, respectively, across 27 genomic datasets. This ensures both robust quantization and efficient fine-tuning in resource-constrained settings.
> 3. **GERM-T for Continual Learning**: Beyond LoRA, GERM-T introduces a novel continual learning strategy that avoids training from scratch while effectively leveraging outlier-free layers. This approach focuses on resource-constrained genomic research, making adaptable fine-tuning without compromising performance possible.
> 4. **Empirical Validation**: Our work provides the first empirical evaluation of integrating outlier mitigation into LoRA fine-tuning and quantization for genomic models, achieving a 64.34% improvement in quantization robustness compared to DNABERT-2. These results validate the effectiveness of our proposed modifications and their impact on genomic tasks.
>
>
> By adapting and extending these methods, we address domain-specific challenges while advancing genomic modeling. We hope this response clarifies the novelty and significance of our contributions, and we are happy to provide further details or analyses if needed.
>
>
> > **Reviewer's Comment**: What is the operational definition …
>
>
> **Response**: We appreciate the opportunity to provide more details about the significance of outliers in transformer-based models. A more detailed explanation is provided in the response to the reviewer `AD8a`.
>
>
> > **Reviewer's Comment**:.Further reference ….
>
>
> **Response**: We appreciate the reviewer's suggestion to expand our literature review by including additional GFMs. In response, we incorporate Evo in the related work section of our revised version.

---

### Decision · Program_Chairs · 2025-05-01

**Decision:**

Accept (poster)

**Comment:**

This submission is proposed to address the limitations of the current GFMs, particularly the DNABERT genomic foundation model. It mainly focuses on the outliers and reduces their impact on the attention calculation. The extensive results show the promise of the proposed methods. More in-depth analysis about the influence of outliers could be very helpful. In general, it is a good paper and we recommend to be accepted.